# Short internal open reading frames repress the translation of N-terminally truncated proteoforms

Raphael Fettig[1,5], Zita Gonda[1,5], Niklas Walter[1,5], Paul Sallmann[1], Christiane Thanisch[1], Markus Winter [1], Susanne Bauer[1], Lei Zhang[2], Greta Linden[2], Margarethe Litfin[1], Marina Khamanaeva[1], Sarah Storm[1], Christina Münzing[1], Christelle Etard[1], Olivier Armant[3], Olalla Vázquez[2,4] & Olivier Kassel [1]✉

## Abstract

**Internal translation initiation sites, as revealed by ribosome profiling experiments can potentially drive the translation of many N-terminally truncated proteoforms. We report that internal short open reading frame (sORF) within coding sequences regulate their translation. nTRIP6 represents a short nuclear proteoform of the cytoplasmic protein TRIP6. We have previously reported that nTRIP6 regulates the dynamics of skeletal muscle progenitor differentiation. Here we show that nTRIP6 is generated by translation initiation at an internal AUG after leaky scanning at the canonical TRIP6 AUG. The translation of nTRIP6 is repressed by an internal sORF immediately upstream of the nTRIP6 AUG. Consistent with this representing a more general regulatory feature, we have identified other internal sORFs which repress the translation of N-terminally truncated proteoforms. In an in vitro model of myogenic differentiation, the expression of nTRIP6 is transiently upregulated through a mechanistic Target of Rapamycin Complex 1-dependent increase in translation initiation at the internal AUG. Thus, the translation of N-terminally truncated proteoforms can be regulated independently of the canonical ORF.**

**Keywords** Myogenesis; Proteoforms; Short ORFs; Translation; Trip6
**Subject Category** Translation & Protein Quality

## Introduction

The number of proteins that can be expressed in a cell far exceeds the number of genes. Alternative splicing and alternative translation of mRNAs drastically increase the coding capacity of the genome by generating multiple proteoforms. How the expression of these various proteoforms is regulated in a physiological context is still partially understood, in particular in the case of alternative translation. In some cases, the proteoforms regulate the function of the "canonical" proteins

(Kelemen et al, 2013; Orre et al, 2019; Bogaert et al, 2020), as for example the case of truncated transcription factors (TFs) which act as dominant-negative inhibitors. However, there are also examples of proteoforms which exert a function distinct from that of their canonical counterparts. This occurs in particular when the canonical protein and its variant are located in different subcellular compartments (Kelemen et al, 2013; Orre et al, 2019; Bogaert et al, 2020). The protein TRIP6 (Lin and Lin, 2011; Willier et al, 2011) represents one such example. TRIP6 belongs to the ZYXIN family of LIM domain-containing proteins that are localized in the cytosol and enriched at sites of cell adhesion where they regulate adhesion and migration. These proteins are also present in the nucleus where they exert transcriptional co-regulator functions for various TFs (Hervy et al, 2006). In the case of TRIP6, this nuclear function is exerted by a shorter, exclusively nuclear proteoform, which we have termed nTRIP6 (Kassel et al, 2004; Diefenbacher et al, 2008, 2010; Kemler et al, 2016). We have recently reported a role for nTRIP6 in the regulation of myogenesis (Norizadeh Abbariki et al, 2021). This process relies on resident adult stem cells, the so-called satellite cells (Schmidt et al, 2019). Upon muscle damage, these cells are activated and give rise to proliferating progenitors, the myoblasts. These then differentiate into committed precursors, the myocytes, which finally fuse together to form multinucleated myofibres. We have shown in an in vitro model of myogenesis that nTRIP6 prevents premature differentiation of myoblasts into myocytes. This repression is required for the subsequent late differentiation and fusion of myocytes. Furthermore, the expression of nTRIP6 is transiently upregulated at the transition between proliferation and differentiation, while that of TRIP6 is not (Norizadeh Abbariki et al, 2021).

Here, we report that nTRIP6 is expressed by alternative translation initiation at an internal AUG and that a short open reading frame (ORF) within the *Trip6* coding sequence represses nTRIP6 translation. Furthermore, the mechanistic target of rapamycin complex 1 (mTORC1) transiently de-represses nTRIP6 translation during early myogenesis. Based on this prototypical example, we identified other mRNAs in which short internal ORFs regulate the translation of N-terminally truncated proteoforms. Thus, our work reveals novel translation cis-regulatory elements that we have termed internal upstream ORFs (iuORFs).

[1]Karlsruhe Institute of Technology (KIT), Institute for Biological and Chemical Systems—Biological Information Processing (IBCS-BIP), Karlsruhe, Germany. [2]Philipps-Universität Marburg, Faculty of Chemistry, Marburg, Germany. [3]Institut de Radioprotection et de Sûreté Nucléaire (IRSN), PSE-ENV/SERPEN/LECO, Cadarache, France. [4]Philipps-Universität Marburg, Center for Synthetic Microbiology (SYNMIKRO), Marburg, Germany. [5]These authors contributed equally: Raphael Fettig, Zita Gonda, Niklas Walter. ✉E-mail: olivier.kassel@kit.edu

# Results

## nTRIP6 is generated by alternative translation at an internal AUG

While the three C-terminal LIM domains of TRIP6 harbor noncanonical nuclear targeting sequences, a nuclear export signal (NES) within its N-terminal pre-LIM region is responsible for its cytosolic localization (Wang and Gilmore 2001). Thus, the 40 kDa nuclear proteoform nTRIP6 should lack a functional NES. However, it should still harbor the three LIM domains as well as the two interaction domains within the pre-LIM region which are required for its transcriptional co-regulator function (Kassel et al, 2004; Diefenbacher et al, 2014; Kemler et al, 2016). Several RefSeq alternative *Trip6* transcripts are described (Fig. EV1), however they do not fulfill these criteria. *Trip6* variant 2 (NM_001417589.1) harbors a frameshift which introduces a premature stop codon. Variant 3 (NM_001417590.1) lacks the NES and harbors an in-frame stop codon in exon 4. The next in-frame initiation codon would generate a predicted protein of 35 kDa. Variant 4 (NR_184539.1) is described as non-coding. It harbors a 52 nucleotides insertion resulting in a premature stop codon upstream of the third LIM domain. We also detected several *Trip6* transcript variants by RT-PCR and 5'-RACE. However, none can encode for a 40 kDa proteoform with the required features (Fig. EV1; Appendix Fig. S1). Splice variant 1 still harbors the NES and lacks both interaction domains; although splice variant 2 lacks the NES, it also lacks the first interaction domain and it encodes a predicted 34 kDa protein; splice variant 3 lacks the 5' part of Exon 4, which leads to a frameshift and a premature stop codon, resulting in a predicted 18 kDa protein; variant 1 identified by 5' RACE lacks exon 2, 3, and part of exon 4, thus lacks the first interaction domain; variant 2 identified by 5' RACE lacks exons 1–3 and part of exon 4, and harbors a putative translation initiation codon which would lead to a 23 kDa protein (Fig. EV1; Appendix Fig. S1). Thus, it seems very unlikely that nTRIP6 is translated from a transcript variant. Transfection of NIH-3T3 fibroblasts with a C-terminally tagged *Trip6* construct gave rise to two major proteins with sizes corresponding to those of TRIP6 and nTRIP6, 55 and 40 kDa, respectively, as well as two other proteins of lower molecular weight (Fig. 1A,B). Deletion of the annotated translation initiation codon (AUG1) abolished the expression of the long proteoform but strongly increased the expression of the 40 kDa proteoform, as well as that of the smaller proteoforms (Fig. 1A–C). This result shows that nTRIP6 is not generated by proteolytic cleavage of TRIP6 and might suggests that it arises from alternative translation initiation after leaky scanning, the process by which the scanning ribosome skips the suboptimal first initiation codon and initiates at a downstream site (Kozak, 2002). Indeed, *Trip6* AUG1 deviates from the optimal Kozak consensus sequence (RNN RNN AUG G, where R represents a purine and N any nucleotide) as it is immediately followed by a conserved T and not by the optimal G (Fig. EV2). We identified a putative initiation codon (AUG2) at position 480 of mouse *Trip6* mRNA (NM_011639.3), in-frame with AUG1 and located in the middle of the NES-encoding sequence. This putative translation initiation site (TIS) is conserved in other species, with either an AUG in a suboptimal Kozak context or, in primates a CUG in an optimal context (Fig. EV2). Several downstream in-frame noncanonical

TISs might be responsible for the translation of the two smaller proteoforms. Mutation of AUG2 into a non-initiating codon abrogated the expression of the 40-kDa proteoform (Fig. 1A–C). To confirm that nTRIP6 is translated at AUG2, we designed a peptide nucleic acid (PNA) to block translation initiation (Marin et al, 2004; Gupta et al, 2017) at this site, as well as a mispaired control PNA (Misp) (Appendix Fig. S2). To validate the effect of the PNA, we designed a reporter construct in which the target sequence encompassing AUG2 was fused in-frame with luciferase, as well as a control construct in which four mismatches were introduced in the target sequence (Fig. EV3A). The AUG2-targeting PNA, but not the mispaired control PNA strongly inhibited the expression of the reporter gene but not that of the mismatch control (Fig. EV3B). In an in vitro transcription/translation assay, the AUG2-targeting PNA inhibited the translation of the 40 kDa proteoform, as compared to the mispaired control PNA, without significantly affecting that of the 55 kDa proteoform (Fig. 1D,E). Deletion of AUG1 abolished the translation of the 55 kDa proteoform and increased the translation of the 40 kDa proteoform, which was inhibited by the AUG2 PNA (Fig. 1D,E), strengthening the hypothesis of leaky scanning at AUG1. We then used the PNA to verify that endogenous nTRIP6 is translated at AUG2. We included a random PNA as an extra control to ensure that the PNA treatment procedure itself does not affect the expression of nTRIP6. Treatment of C2C12 myoblasts with the AUG2-targeting PNA reduced the expression of nTRIP6 without significantly affecting that of the long proteoform TRIP6 (Fig. 1F,G). Importantly, treatment with the AUG2 PNA did not decrease, but rather increased *Trip6* mRNA levels in C2C12 myoblasts (Fig. EV3C). Together, these results show that nTRIP6 is generated by translation initiation at AUG2, presumably after leaky scanning at AUG1.

The introduction of a hairpin immediately after the transcription initiation site in the tagged *Trip6* construct (Fig. 2A) to inhibit cap-dependent translation (Kozak, 1989) strongly reduced the expression of both TRIP6 and nTRIP6 as compared to a non-structured sequence (Fig. 2B,C). The expression of TRIP6 and of nTRIP6 were reduced to the same extent, as shown by their expression ratio (Fig. 2D). This reduced expression was not the consequence of a decrease in the mRNA level of the hairpin-containing construct. Indeed, the translation of the eGFP used as a transfection/loading control, which was driven by an IRES in the same mRNA was not decreased by the presence of the hairpin (Fig. 2B). This result confirms that the translation of nTRIP6 is cap-dependent and not mediated by an internal ribosome entry site. Mutating the AUG1 TIS into a "perfect" Kozak sequence (Kozak, 2002) did not significantly reduce the expression of TRIP6 and nTRIP6 (Fig. 2E–G). However, weakening the AUG1 TIS decreased the expression of TRIP6 and strongly increased that of nTRIP6 (Fig. 2E–G). Together, these results show that nTRIP6 arises from translation initiation at AUG2 after leaky scanning at AUG1.

## nTRIP6 translation transiently increases during in vitro myogenesis

Our previous observation that the levels of nTRIP6 transiently increased during C2C12 myoblast differentiation, whereas those of TRIP6 did not vary (Norizadeh Abbariki et al, 2021) suggests an increased translation

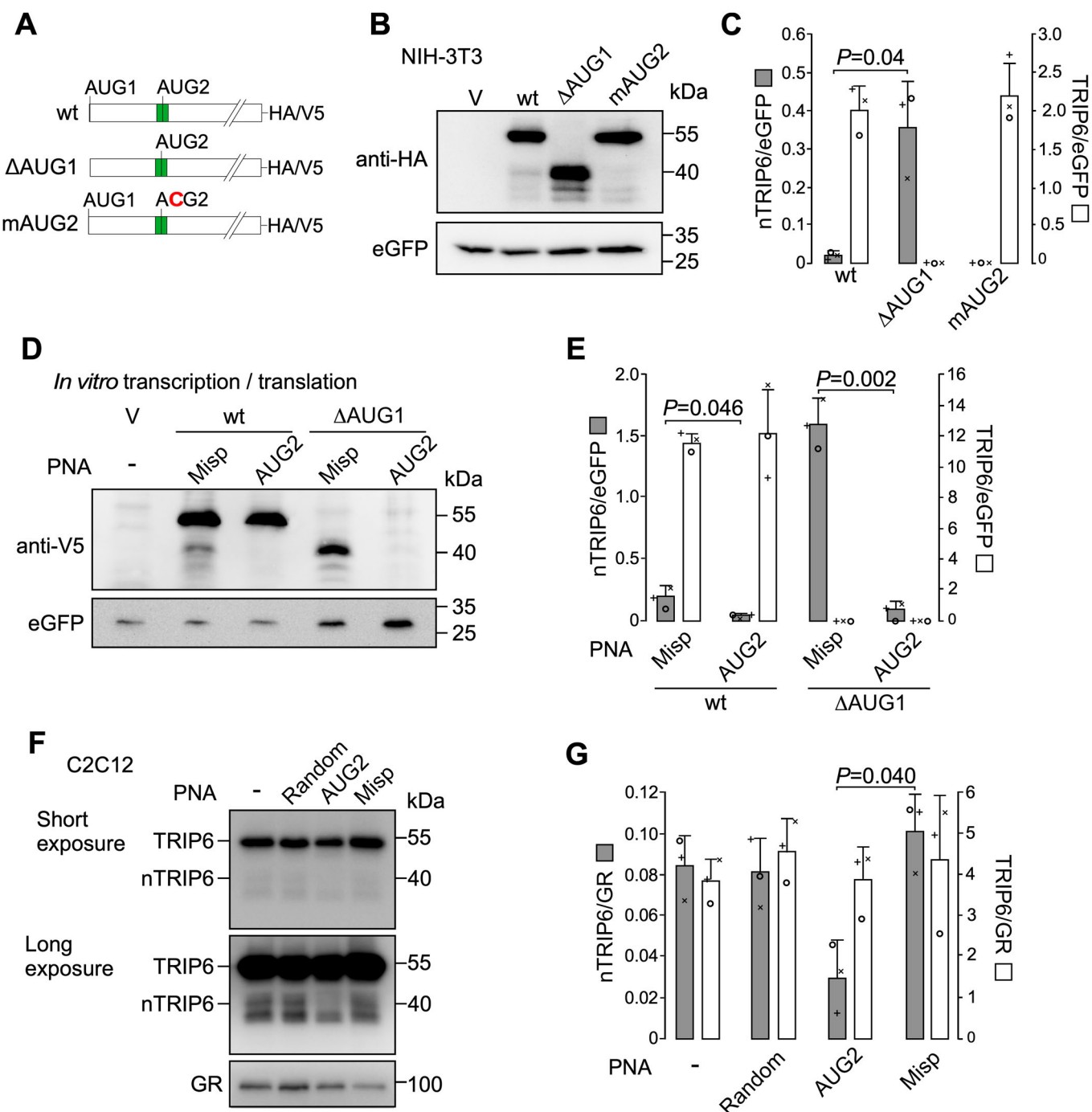

**Figure 1. nTRIP6 is generated by alternative translation at an internal AUG.**

(A) Schematic representation of the constructs used. The green box represents the Nuclear Export Signal encoding sequence. (B, C) Lysates of NIH-3T3 fibroblasts transfected with either an empty vector (V) or the indicated HA-tagged TRIP6 constructs, together with eGFP as a transfection control were subjected to western blotting using antibodies against the HA tag and eGFP. Representative blots are shown (B). The expression of TRIP6 and nTRIP6 relative to the transfection control is presented as mean ± SD of three independent experiments (C). (D, E) In vitro transcription and translation of V5-tagged *Trip6* CDS constructs with (wt) or without the first AUG (ΔAUG1), as well as of eGFP as in internal translation control, in the presence of a peptide nucleic acid (PNA) targeting the second AUG (AUG2) or a mispaired control PNA (Misp), and analysis by western blotting using anti-V5 and anti-eGFP antibodies. Representative blots are shown (D). The expression of TRIP6 and nTRIP6 relative to the eGFP control is presented as mean ± SD of three independent experiments (E). (F, G) Lysates of C2C12 myoblasts either mock-treated (−), treated with a cell-penetrating random PNA, the AUG2 or the Misp PNA were subjected to western blotting using antibodies against TRIP6/nTRIP6 and GR as a loading control. Representative blots are shown (F). The expression of TRIP6 and nTRIP6 relative to the loading control is presented as mean ± SD of three independent experiments (G). In each graph, individual values are depicted by symbols, each representing an independent experiment. Student's paired *t* test *P* values are indicated. Source data are available online for this figure.

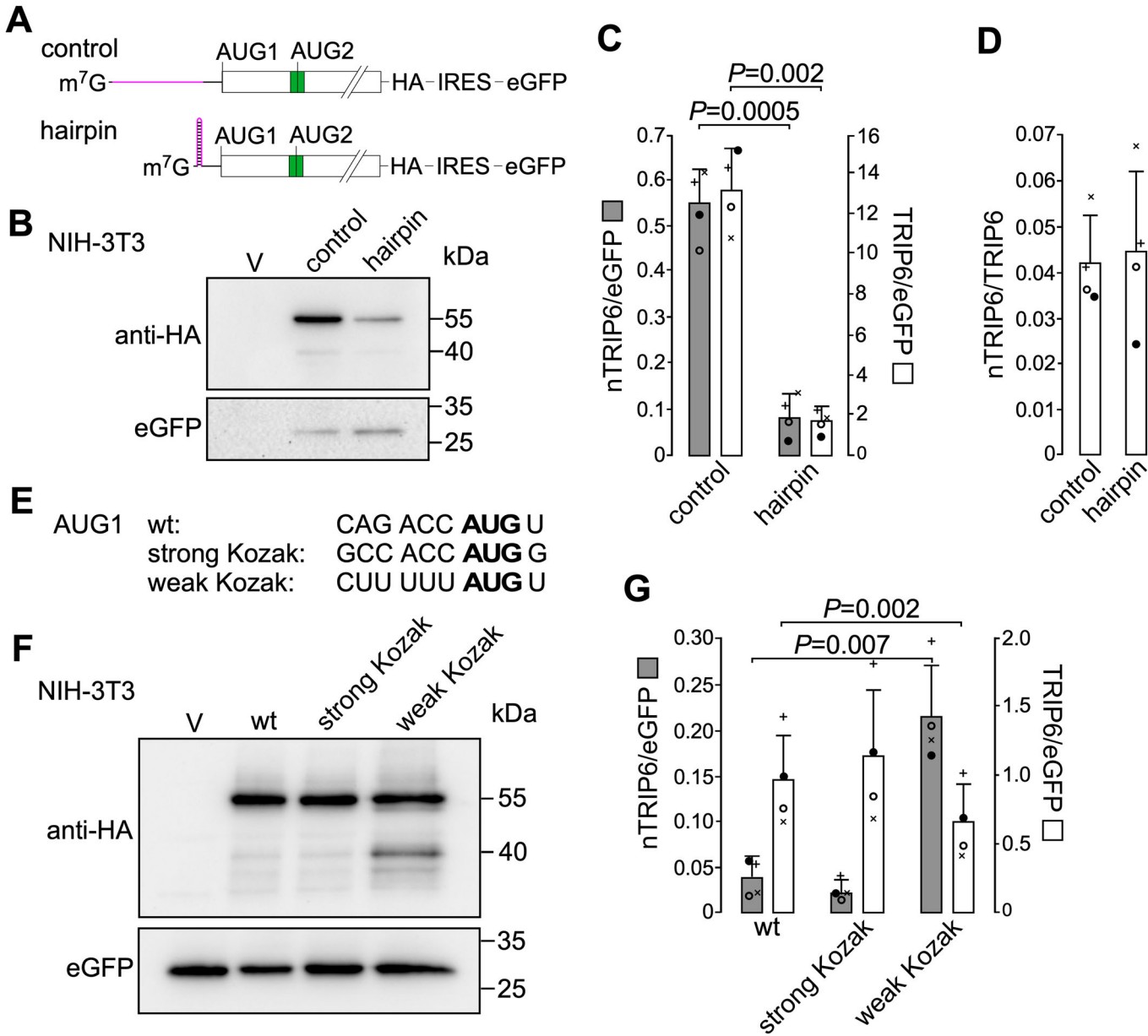

**Figure 2. nTRIP6 is translated after leaky scanning at AUG1.**

(A) Schematic representation of the constructs used. The green box represents the Nuclear Export Signal encoding sequence. (B–D) Lysates of NIH-3T3 fibroblasts transfected with either an empty vector (V) or the indicated HA-tagged TRIP6 constructs, together with eGFP as a transfection control were subjected to western blotting using antibodies against the HA tag and eGFP. Representative blots are shown (B). The expression of TRIP6 and nTRIP6 relative to the transfection control (C) and the expression of nTRIP6 relative to that of TRIP6 (D) are presented as mean ± SD of three independent experiments. (E) The sequence of the AUG1 translation initiation sites in the constructs used in (F, G) (wt, wild-type sequence). (F, G) Lysates of NIH-3T3 fibroblasts transfected with either an empty vector (V) or the indicated HA-tagged TRIP6 constructs, together with eGFP as a transfection control were subjected to western blotting using antibodies against the HA tag and eGFP. Representative blots are shown (F). The expression of TRIP6 and nTRIP6 relative to the transfection control is presented as mean ± SD of three independent experiments. In each graph, individual values are depicted by symbols, each representing an independent experiment. Student's paired *t* test *P* values are indicated. Source data are available online for this figure.

at AUG2. To address this question, we used the AUG2 and the control PNAs fused to a cell-penetrating peptide in a C2C12 differentiation assay. In the control PNA-treated cells, nTRIP6 expression transiently increased during the proliferation phase while TRIP6 levels did not significantly vary. The PNA targeting AUG2 abolished the increase in nTRIP6 expression without significantly affecting the levels of TRIP6 (Fig. 3A,B). Surprisingly, in cells treated with the PNA targeting AUG2 a

new proteoform was visible on the western blot just below the TRIP6 band (Fig. 3A). This proteoform most likely arises from a CUG initiation site in a relatively good Kozak context located 78 nucleotides downstream of AUG1. Furthermore, we quantified AUG1 and AUG2 translation initiation sites (TIS1 and TIS2) amongst ribosome-protected mRNA fragments in C2C12 cells. Cells were treated with harringtonine to immobilize initiating ribosomes (Ingolia et al, 2012) during the

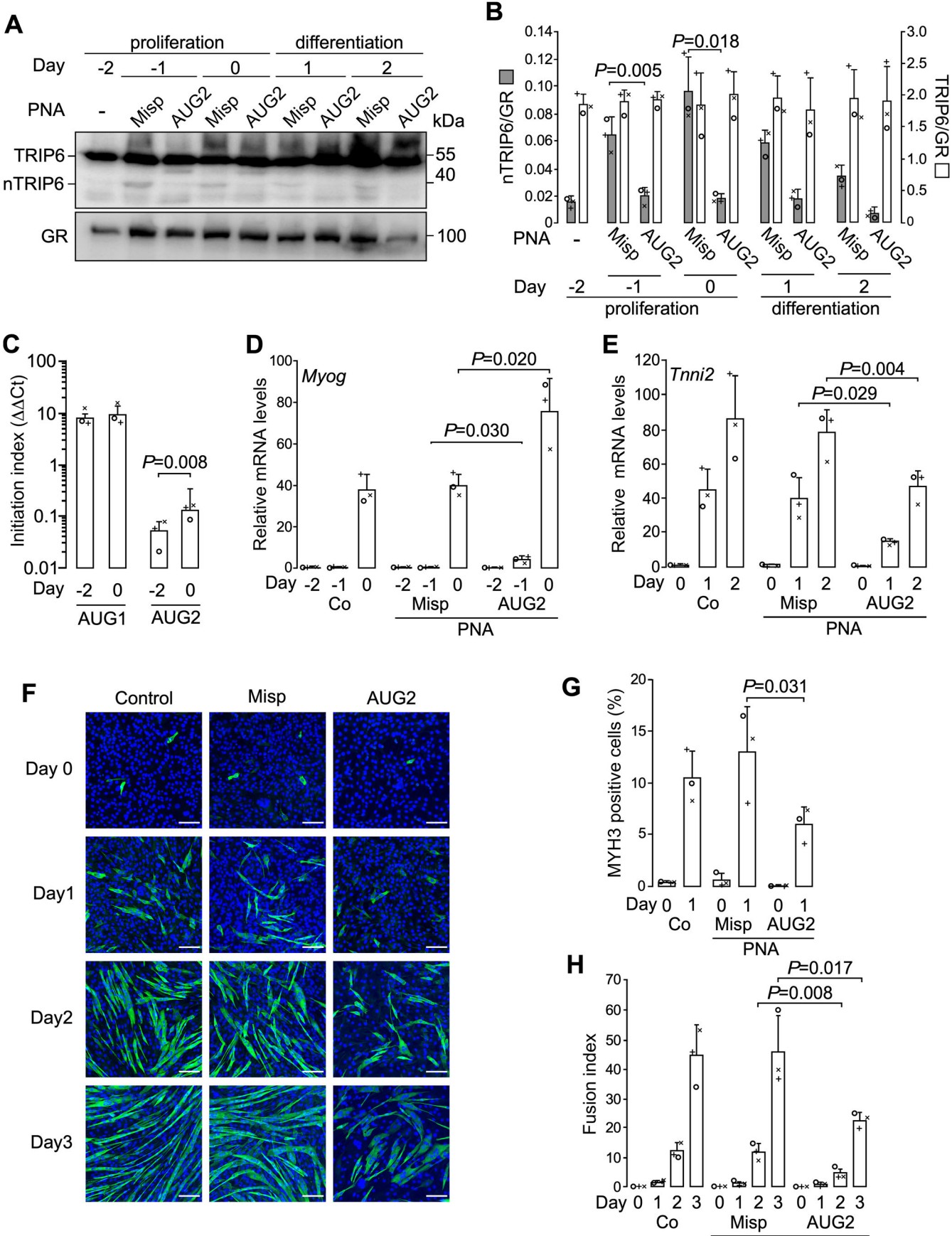

◀ **Figure 3. nTRIP6 translation transiently increases during in vitro myogenesis.**

(A, B) C2C12 myoblasts were treated with either the cell-penetrating AUG2 or Misp PNAs, harvested at the indicated time point of a differentiation experiment and subjected to western blotting using antibodies against TRIP6/nTRIP6 and GR as a loading control. Representative blots are shown (A). The expression of TRIP6 and nTRIP6 relative to the loading control is presented as mean ± SD of three independent experiments (B). (C) C2C12 myoblasts were treated at the indicated day of a differentiation experiment with Harringtonine to arrest 80 s ribosomes immediately after translation initiation. Initiation at AUG1 and AUG2 was determined by reverse transcription and real-time PCR analysis of the respective ribosome-protected RNA fragments. The initiation index was obtained after correction for the background signal detected at a non-initiating site on *Trip6* mRNA and normalization to the initiation at the AUG of glucocorticoid receptor (GR) mRNA after background correction (non-initiating site on GR mRNA) by the ΔΔCt method. Results are presented as mean ± SD of three independent experiments. (D–H) Mock-treated control C2C12 cells (Co) or cells treated with either the AUG2 or the Misp PNAs were subjected to a differentiation experiment. (D, E) The relative levels of the *Myog* (D) and *Tnni2* (E) mRNAs were determined by reverse transcription and real-time PCR. Results are plotted relative to the expression of the *Rplp0* gene (mean ± SD of three independent experiments). F-H Cells were fixed at the indicated day and subjected to immunofluorescence analysis using an antibody against MYH3. Nuclei were counterstained using DAPI. (F) Representative images are presented (scale bar 200 μm). The percentage of MYH3 expressing mononuclear cells (G) and the fusion index (percentage of nuclei within fused myotubes) (H) are presented as mean ± SD of three independent experiments. In each graph, individual values are depicted by symbols, each representing an independent experiment. Student's paired *t* test *P* values are indicated. Source data are available online for this figure.

**Table 1. Kinases which increase the expression of nTRIP6.**

| Kinase | nTRIP6/TRIP6 | Z-score[a] | *P* value |
|---|---|---|---|
| Mark1 | 0.0964 | 3.016 | $2.56 \times 10^{-3}$ |
| Map3k7 | 0.0896 | 2.578 | $9.94 \times 10^{-3}$ |
| Eif2ak1 | 0.0880 | 2.471 | $1.35 \times 10^{-2}$ |
| Cdc7 | 0.0875 | 2.443 | $1.71 \times 10^{-2}$ |
| Chuk | 0.0866 | 2.385 | $1.71 \times 10^{-2}$ |
| mTor | 0.0847 | 2.258 | $2.39 \times 10^{-2}$ |
| Akt1 | 0.0832 | 2.163 | $3.05 \times 10^{-2}$ |
| Pkn2 | 0.0804 | 1.983 | $4.74 \times 10^{-2}$ |
| Vector[b] | 0.0371 ± 0.015 | | |

[a] Z scores and derived *P* values were determined for the entire kinase set (see "Methods").
[b] The nTRIP6/TRIP6 ratio in cells transfected with the empty vector is presented a mean ± SD of 16 determinations.

proliferation phase (day −2), when nTRIP6 levels are low, and at the start of differentiation (day 0) when nTRIP6 levels are increased. TIS1 was detected at the highest level (Fig. 3C), confirming that most of the initiation occurs at AUG1, and this level did not vary significantly between day −2 and day 0. TIS2 was also detected, although at much lower levels, confirming translation initiation at AUG2. Furthermore, the level of initiating ribosome-protected TIS2 significantly increased between day −2 and day 0 (Fig. 3C). Thus, the transient increase in nTRIP6 expression occurs via increased translation initiation at AUG2. We have previously shown that nTRIP6 represses premature myoblast differentiation, allowing proper myocyte differentiation and fusion at later stages (Norizadeh Abbariki et al, 2021). Accordingly, treatment of C2C12 myoblasts with the AUG2 PNA accelerated the expression of *Myog* mRNA, used as an indicator of early myocytic differentiation (Edmondson and Olson, 1989), and delayed the expression of *Tnni2*, a late differentiation gene (Lin et al, 1994), as compared to untreated cells or cells treated with the control PNA (Fig. 3D,E). Similarly, during the early myocytic differentiation phase (day 1 after the induction of differentiation), the AUG2 PNA reduced the number of cells expressing MYH3 (embryonic myosin heavy chain), another index of late differentiation (Fig. 3F,G). Finally, cell fusion, which started at day 2 and strongly increased at day 3 in untreated cells and in cells treated with the control PNA, was significantly inhibited in cells treated with the AUG2 PNA (Fig. 1F,H).

## The mTOR pathway transiently stimulates nTRIP6 translation during myogenesis

In order to identify signaling pathways involved in the upregulation of nTRIP6 translation, we screened a library of 184 unique Medaka kinases (Chen et al, 2014; Souren et al, 2009) for those altering the nTRIP6/TRIP6 ratio. This ratio was normally distributed (Fig. EV4A,B), and analysis of the Z scores (Fig. EV4C; Table EV1) revealed 8 kinases which significantly increased it (Table 1), including notably mTOR. Interestingly, AKT1, CHUK and MAP3K7, other hits from the screen, are direct or indirect activators of mTORC1 (Saxton and Sabatini, 2017). We thus tested the effect of mTORC1 on nTRIP6 expression. In proliferating C2C12 myoblasts, treatment with IGF-1, a potent inducer of mTORC1 (Saxton and Sabatini, 2017) increased the phosphorylation of the mTORC1 substrate p70S6K (Fig. 4A,C). IGF-1 treatment did not significantly affect TRIP6 levels. However, it increased the expression of nTRIP6. This increase was partially inhibited by the mTORC1 inhibitor Everolimus (Fig. 4A,B), confirming the involvement of mTORC1. Furthermore, a constitutively active mutant of RHEB (RHEB-Q64L) (Jiang and Vogt, 2008), the immediate upstream activator of mTORC1, increased the expression of nTRIP6 from the tagged construct without significantly affecting the expression of TRIP6. This increase was inhibited by Everolimus (Fig. 4D,E), confirming the involvement of mTORC1. Thus, nTRIP6 translation is stimulated by the mTORC1 pathway. We next investigated whether mTORC1 was responsible for the upregulation of nTRIP6 translation during C2C12 cell differentiation. mTORC1 activity, as assessed by the phosphorylation of p70S6K was elevated in the proliferation phase, reaching a maximum at day −1 (relative to the induction of differentiation), and then decreased prior to differentiation (Fig. 4F,G). Treatment with Everolimus strongly reduced the transient increase in nTRIP6 (Fig. 4H,I). Together, these results show that during early myogenesis the transient increase in mTORC1 activity promotes the increase in nTRIP6 translation.

## A short internal upstream open reading frame (iuORF) represses nTRIP6 translation

The stimulatory effect of mTORC1 on nTRIP6 translation is reminiscent of the regulation of translation by short upstream ORFs (uORFs) in the 5' regulatory regions of mRNAs which

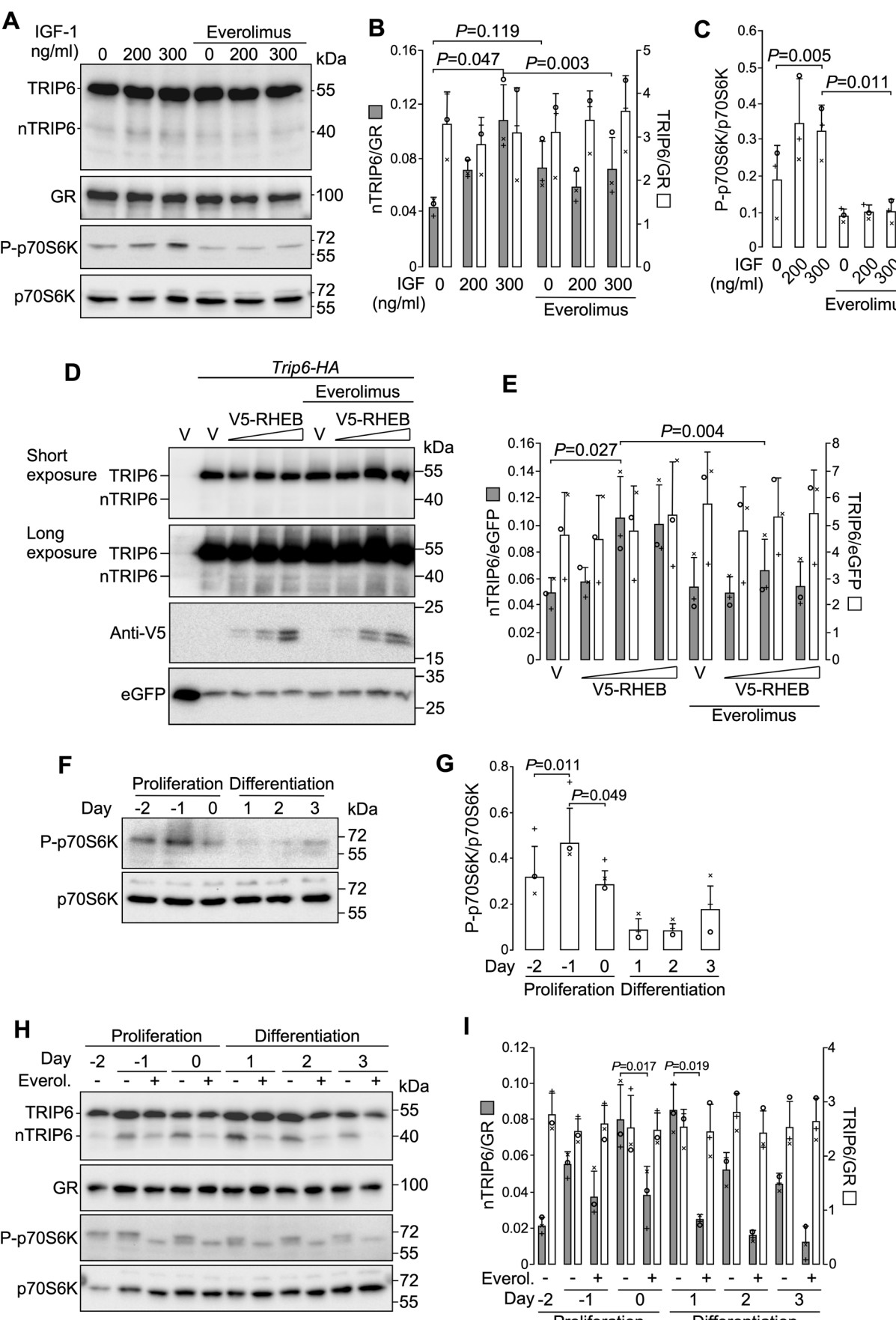

**Figure 4. mTORC1 mediates the increase in nTRIP6 translation during myogenesis.**

(A–C) Lysates of C2C12 cells treated for 6 h with IGF-I at the indicated concentration in the presence of either solvent alone or the mTORC1 inhibitor Everolimus (100 nM) were analyzed by western blotting using antibodies against TRIP6/nTRIP6, phosphorylated (Thr389) p70 S6 Kinase (P-p70S6K) and GR as a loading control. Membranes were stripped and re-probed with a phosphorylation-independent p70S6K antibody. Representative western blots are shown (A). The expression of TRIP6 and nTRIP6 relative to the loading control (B) and the amount of P-p70S6K normalized to the total amount of p70S6K (C) are presented as mean ± SD of three independent experiments. (D, E) C2C12 cells were transfected with the HA-tagged Trip6 construct together with either an empty vector (V) or a V5-tagged constitutively active mutant of RHEB (RHEB-Q64L) at increasing doses, together with eGFP as a transfection control. Cells were treated 16 h later with solvent alone or Everolimus (50 mM) and harvested 6 h later. Lysates were subjected to western blotting using antibodies against the HA tag, the V5 tag and eGFP. Representative western blots are shown (D). The expression of nTRIP6 and TRIP6 relative to the transfection control is presented as mean ± SD of three independent experiments. (F, G) Lysates of C2C12 myoblasts harvested at the indicated day of a differentiation experiment were subjected to western blotting using an anti-P-p70S6K. Membranes were stripped and re-probed with a phosphorylation-independent p70S6K antibody. Representative blots are shown (F). The amount of P-p70S6K normalized to the total amount of p70S6K is presented as mean ± SD of three independent experiments (G). (H, I) Lysates of C2C12 myoblasts treated as indicated with either solvent alone or Everolimus (Everol.; 100 nM) and harvested at the indicated day of a differentiation experiment were subjected to western blotting using antibodies against TRIP6/nTRIP6, P-p70S6K and GR as a loading control. Membranes were stripped and re-probed with a phosphorylation-independent p70S6K antibody. Representative blots are shown (H). The expression of TRIP6 and nTRIP6 relative to the loading control is presented as mean ± SD of three independent experiments (I). In each graph, individual values are depicted by symbols, each representing an independent experiment. Student's paired t test P values are indicated. Source data are available online for this figure.

typically repress translation initiation at the downstream main ORF TIS (Chen and Tarn, 2019; Zhang et al, 2019). Indeed, apart from its role in stimulating translation initiation of a subset of mRNAs (Masvidal et al, 2017), mTORC1 stimulates reinitiation downstream of uORFs (Calkhoven et al, 2000; Zidek et al, 2015; Chen et al, 2010). We identified a conserved short ORF in the Trip6 mRNA-coding sequence (Figs. 5A and EV5) located immediately upstream of AUG2, at position 359–466 of mouse Trip6 mRNA. In the case of uORFs in 5' regulatory regions, the sequence context of the initiation codon, the length of the short ORF, and its distance to the next downstream TIS are the most relevant features to repress initiation at this downstream TIS (Chen and Tarn, 2019; Zhang et al, 2019). The initiation codon of the short ORF in Trip6 mRNA is in a favorable context for initiation (G in position +4). Furthermore, the short ORF is relatively long (78 nucleotides) and the distance between the stop of the short ORF to AUG2 is rather short (14 nucleotides). These features are very conserved (Fig. EV5) and predictive of a repressive effect on the translation of nTRIP6. Mutation of the short ORF initiation codon into a non-initiating codon in the tagged TRIP6 construct led to an increased expression of nTRIP6 (Fig. 5A–C). Furthermore, expression of nTRIP6 from the AUG1-deleted construct was further increased upon mutation of the short ORF AUG (Fig. 5A–C), confirming the repressive role of the short ORF for initiation at AUG2 after leaky scanning through AUG1. The very low levels of nTRIP6 detected in control conditions show that the short ORF does not fully repress translation initiation at AUG2. This suggests leaky scanning at the short ORF AUG, which is in a strong yet imperfect Kozak context (CCC AUG G). Weakening the Kozak context of the short ORF TIS increased the expression of nTRIP6. Furthermore, nTRIP6 expression was further increased when both AUG1 and the short ORF AUG were in a weak Kozak context (Fig. 5D–F). Thus, leaky scanning at AUG1 together with either leaky scanning at the short ORF AUG or reinitiation after translation of the short ORF enable weak nTRIP6 translation. Repression of translation by uORFs in 5' leader sequences depends on the intercistronic distance between the uORF and the main ORF, a short distance preventing reinitiation by post-termination ribosomes (Chen and Tarn, 2019; Zhang et al, 2019). Reducing the distance between the short ORF stop codon and AUG2 from 14 to 5 nucleotides had no significant effect on the expression of nTRIP6. However, increasing

it to 50 nucleotides strongly increased the expression of nTRIP6 without affecting that of TRIP6 (Fig. 5G–I). Thus, the short ORF prevents reinitiation at AUG2. Given the similarities between this short ORF and "classical" repressive uORFs in 5' leader sequences, we designated this regulatory element an internal uORF (iuORF).

mTORC1 has been shown to inhibit post-termination ribosome recycling after the translation of uORFs in 5' leader sequences, and thereby to promote reinitiation at downstream start codons (Schepetilnikov et al, 2013; Gunišová et al, 2018; Zidek et al, 2015). We therefore investigated whether mTORC1 stimulates nTRIP6 translation by a similar mechanism. If so, the stimulatory effect of mTORC1 should depend on the distance between the iuORF stop codon and AUG2. The constitutively active RHEB mutant, which increased nTRIP6 expression from the wild-type construct, did not increase it when the intercistronic distance was reduced to 5 nucleotides or increased to 50 nucleotides (Fig. 6A,B). Furthermore, the ribosome protection assay in harringtonine-treated C2C12 myoblasts showed that the increased translation initiation at TIS2 between day −2 and day 0 of differentiation was inhibited by Everolimus. Conversely, initiation at TIS1 and at the iuORF TIS was not significantly modified between day −2 and day 0 and not affected by Everolimus (Fig. 6C). Together, these results indicate that mTORC1 stimulates reinitiation at AUG2.

To investigate whether other mRNAs also harbor iuORFs, we curated a published ribosome profiling dataset that identified over 13,000 TISs in mouse embryonic stem cells (Ingolia et al, 2011b; Data ref: Ingolia et al, 2011a). Based on stringent filtration criteria (see Methods section), among the 4994 genes analyzed, we identified 20 mRNAs exhibiting a short internal ORF in another frame than the canonical ORF and in close proximity to a TIS for an N-terminally truncated proteoform. Among these short internal ORFs 5 were upstream, with the stop codon immediately upstream of the truncated proteoform TIS (Fig. 7A), and 15 were overlapping, with the stop codon downstream of the truncated proteoform TIS (Fig. 7B). The sequence context of the initiation codon, the length of the short ORF and its distance to the N-terminally truncated proteoform TIS were highly conserved in the human and rat mRNAs (Table EV2). We validated this finding by studying the contribution of the identified TISs in Kctd20 mRNA, which bears an upstream iuORF and in Tfdp1 mRNA which harbors an overlapping iuORF (Fig. 8). Transfection of

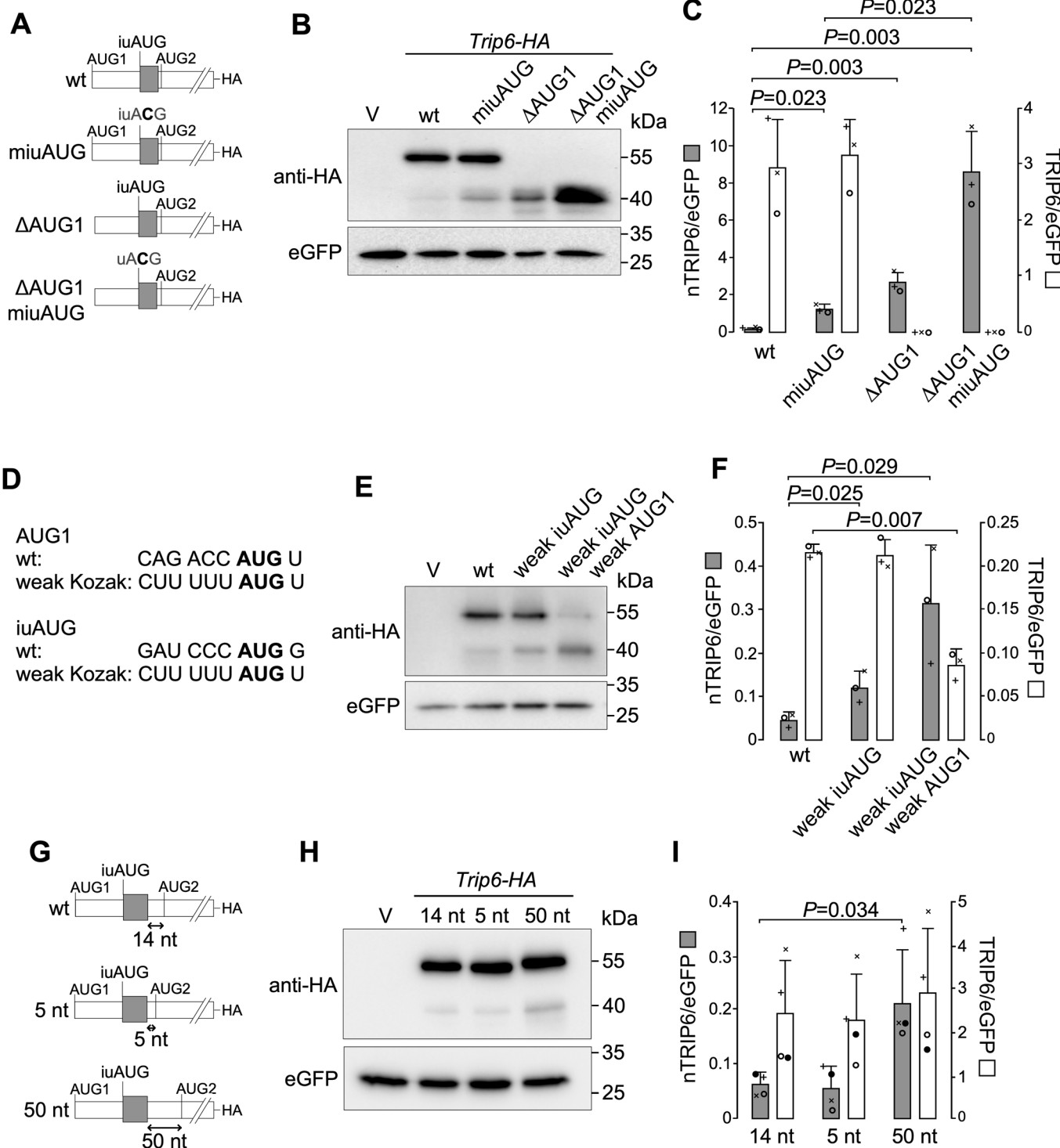

NIH-3T3 cells with a C-terminally tagged KCTD20 construct gave rise to a protein with a size corresponding to the usage of the annotated first AUG, which was indeed not expressed when this AUG was mutated into a non-initiating codon. A slightly smaller protein, presumably translated from a downstream noncanonical TIS was also detected, which was still present when AUG1 was mutated. The wild-type construct also gave rise to a short

proteoform with a size corresponding to the predicted size of the postulated N-terminally truncated proteoform. Its expression was strongly increased when the first AUG was mutated, and was abolished when the AUG for the postulated N-terminally truncated proteoform (AUG2) was mutated into a non-initiating codon (Fig. 8A–C). These results suggest that this short proteoform is translated from AUG2 after leaky scanning at AUG1. Furthermore,

**Figure 5.  A short internal ORF represses the translation of nTRIP6.**

(A) Schematic representation of the TRIP6 constructs used. The internal upstream open reading frame (iuORF) is indicated as a gray box. (B, C) Lysates of NIH-3T3 fibroblasts transfected with either an empty vector (V) or the indicated constructs, together with eGFP as a transfection control were analyzed by western blotting using anti-HA and anti-eGFP antibodies. Representative blots are shown (B). The expression of TRIP6 and nTRIP6 relative to the transfection control is presented as mean ± SD of three independent experiments (C). (D) Sequences of the wild-type (wt) and mutated (weak Kozak) AUG1 and iuORF AUG (iuAUG) translation initiation sites. (E, F) Lysates of NIH-3T3 fibroblasts transfected with either an empty vector (V) or the indicated constructs, together with eGFP as a transfection control were analyzed by western blotting using anti-HA and anti-eGFP antibodies. Representative blots are shown (E). The expression of TRIP6 and nTRIP6 relative to the transfection control is presented as mean ± SD of three independent experiments (F). (G) Schematic representation of the TRIP6 constructs used, in which the spacer between the stop codon of the iuORF and AUG2, 14 nucleotides (nt) in wt *Trip6*, is either reduced to 5 nt or increased to 50 nt. (H, I) Lysates of NIH-3T3 fibroblasts transfected with either an empty vector (V) or the indicated constructs, together with eGFP as a transfection control were analyzed by western blotting using anti-HA and anti-eGFP antibodies. Representative blots are shown (H). The expression of TRIP6 and nTRIP6 relative to the transfection control is presented as mean ± SD of three independent experiments (F). In each graph, individual values are depicted by symbols, each representing an independent experiment. Student's paired *t* test *P* values (FDR-corrected in (C, F)) are indicated. Source data are available online for this figure.

mutation of the identified upstream iuORF initiation codon into a non-initiating codon led to an increased expression of the short proteoform (Fig. 8A–C). Similarly, a C-terminally tagged TFDP1 construct gave rise to a protein with a size corresponding to the usage of the annotated first AUG. Its expression was strongly reduced but not totally abolished when this AUG was mutated into a non-initiating codon (Fig. 8D–F), suggesting that in the context of this mRNA the ACG codon used worked as a weak initiation codon, as already reported (Ingolia et al, 2011b). A shorter proteoform with a size corresponding to that of the postulated N-terminally truncated proteoform was expressed at very low levels from the wild-type construct. Its expression was slightly increased from the construct with a mutated AUG1, and strongly increased when the identified overlapping iuORF initiation codon was mutated into a non-initiating codon (Fig. 8D–F). Furthermore, similarly to what we observed with nTRIP6, transfection of the constitutively active mTORC1 activator RHEB-Q64L led to an increase in the expression of the short KCTD20 proteoform from the tagged construct, whereas the modest increase in the expression of canonical long proteoform was not significant (Fig. 8G,H). However, neither the short nor the canonical long proteoforms of TFDP1 were regulated by RHEB-Q64L (Fig. 8I,J).

## Discussion

Using *Trip6* mRNA as a model, we have identified short internal ORFs as novel cis-regulatory elements which regulate the translation of N-terminally truncated proteoforms.

We show that nTRIP6 is translated from an internal AUG located in the middle of the NES-encoding sequence. Given that the NES is the main determinant of the cytosolic localization of TRIP6 (Wang and Gilmore, 2001), the nuclear localization of nTRIP6 (Kassel et al, 2004; Kemler et al, 2016) is a logical consequence of the truncated NES. Our results show that nTRIP6 translation occurs after leaky ribosome scanning at the first AUG responsible for TRIP6 translation. Leaky scanning occurs when the first initiation codon is in a suboptimal context (Kozak, 2002). Indeed, the *Trip6* AUG1 is immediately followed by a conserved T and not by the optimal G. However, leaky scanning at AUG1 is not very prominent. Our analysis of the ribosome-protected translation initiation sites revealed that most of the translation initiation occurs at this first AUG. The relative initiation rates at the first and second AUGs are consistent with the very low levels of nTRIP6

relative to TRIP6 in all cells tested (Kassel et al, 2004; Diefenbacher et al, 2008, 2010; Kemler et al, 2016). Furthermore, the difference in the relative levels of the two proteoforms reflects their functions. While TRIP6 is a regulator of the cytoskeleton, nTRIP6 serves as a transcriptional regulator. These are generally expressed at significantly lower levels than cytoskeletal regulatory proteins (Beck et al, 2011).

Ribosome profiling experiments have revealed that many N-terminally truncated proteoforms are generated by internal translation initiation (Ingolia et al, 2011b; Damme et al, 2014). Here, we show that the translation of such proteoforms can be regulated. Indeed, nTRIP6 translation is repressed by the short ORF that we have identified in the coding sequence. Given its similarities with the repressive uORFs described in the 5' regulatory regions of many mRNAs (Chen and Tarn, 2019; Zhang et al, 2019), we have termed this novel regulatory element an internal uORF (iuORF). The first hint that this iuORF might be functional came from the observation that the sequence context of the iuORF AUG, its length and its distance from the nTRIP6 AUG, which are most relevant for uORFs to repress initiation at downstream AUGs (Chen and Tarn, 2019; Zhang et al, 2019), were highly conserved. Conversely, the iuORF coding sequence was not conserved, pointing to an absence of function of the encoded peptide. We confirmed the functionality of the iuORF by showing that mutating the iuAUG into a non-initiating codon increased nTRIP6 expression. Thus, the repressive iuORF in *Trip6* mRNA provides a second mechanism that limits nTRIP6 translation. However, this repression is not total since nTRIP6 expression is detected in control conditions. This low level of translation initiation at AUG2 could occur either by leaky scanning at the iuORF TIS or by reinitiation after translation of the iuORF. The rather long iuORF and the short distance to AUG2 is predictive of a very inefficient reinitiation. Our results showing that decreasing the intercistronic distance has no effect on nTRIP6 expression confirms that reinitiation after the iuORF is already maximally inhibited in the wild-type situation. Conversely, increasing this distance strongly increased nTRIP6 expression. Thus, the low levels of nTRIP6 translation most likely arises after leaky scanning at AUG1 and the iuORF TIS.

The existence of short ORFs in the coding sequences of mRNAs has been revealed by ribosome profiling experiments (Ingolia et al, 2011b; Bazzini et al, 2014). However, their function has remained elusive. Our analysis of these short ORFs in a published ribosome profiling dataset (Ingolia et al, 2011b; Data ref: Ingolia et al, 2011a)

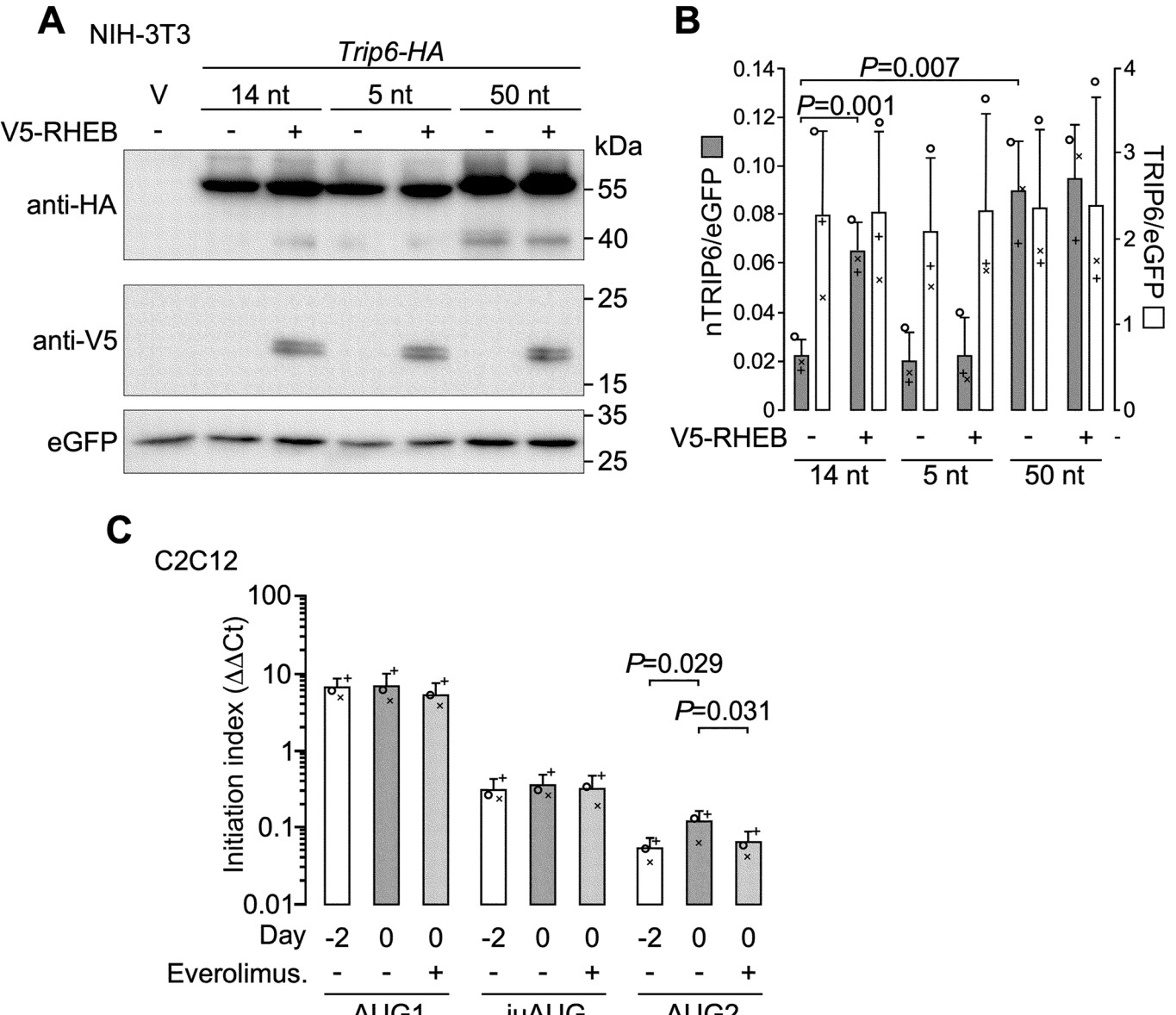

**Figure 6.  mTORC1 stimulates nTRIP6 translation by favoring reinitiation after translation of the iuORF.**

(A, B) NIH-3T3 fibroblasts were transfected with either an empty vector (V) or constructs harboring spacers of variable length between the stop codon of the iuORF and AUG2 (nt, nucleotide; see Fig. 5G legend), together with either an empty vector (V) or a V5-tagged constitutively active mutant of RHEB (RHEB-Q64L) and eGFP as a transfection control. Lysates were analyzed by western blotting using anti-HA, anti-V5, and anti-eGFP antibodies. Representative blots are shown (A). The expression of TRIP6 and nTRIP6 relative to the transfection control is presented as mean ± SD of three independent experiments (B). (C) C2C12 myoblasts were subjected to a differentiation experiment and treated at day −2 with either solvent alone or the mTORC1 inhibitor Everolimus (100 nM) as indicated. Cells were then treated at the indicated day with Harringtonine to arrests 80 s ribosomes immediately after translation initiation. Initiation at AUG1, iuORF AUG (iuAUG) and AUG2 was determined by reverse transcription and real-time PCR of the respective ribosome-protected RNA fragments. The initiation index was obtained after correction for the background signal detected at a non-initiating site on *Trip6* mRNA and normalization to the initiation at the AUG of glucocorticoid receptor (GR) mRNA after background correction (non-initiating site on GR mRNA) by the ΔΔCt method. Results are presented as mean ± SD of three independent experiments. In each graph, individual values are depicted by symbols, each representing an independent experiment. Student's paired *t* test *P* values are indicated. Source data are available online for this figure.

led to the identification of other iuORFs which regulate the translation of N-terminally truncated proteoforms. In this analysis of 4,994 mRNAs, we identified only 20 potential iuORFs for which, as in the case of the *Trip6* mRNA iuORF, the features most relevant for a repressive effect were highly conserved. Using *Kctd20* and *Tfdp1* mRNAs as examples of mRNAs bearing an upstream and an overlapping iuORF, respectively, we confirmed that iuORFs do repress the translation of N-terminally truncated proteoforms. However, we cannot completely exclude the possibility that the short proteoforms are translated from alternative mRNAs lacking the first AUG, in which case the short ORF would rather represent a "classical" uORF.

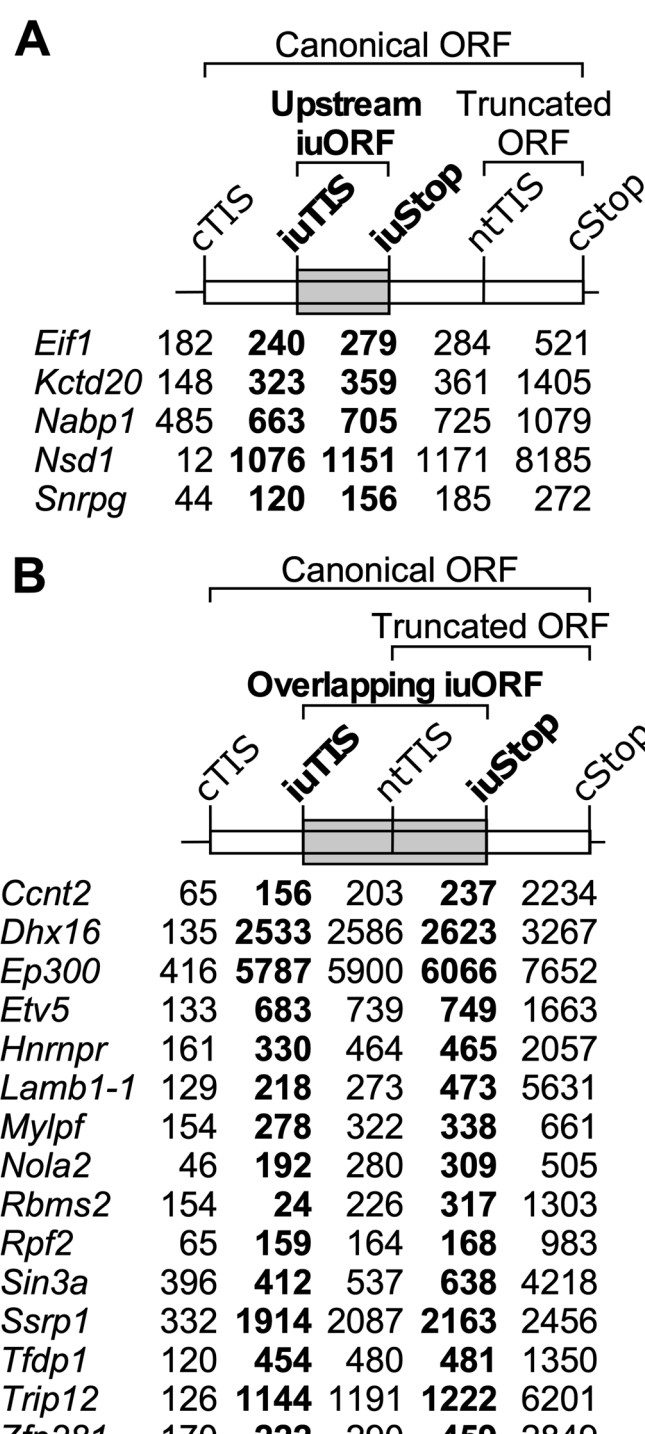

**A**

Canonical ORF

Upstream iuORF Truncated ORF

cTIS iuTIS iuStop ntTIS cStop

| | cTIS | iuTIS | iuStop | ntTIS | cStop |
|---|---|---|---|---|---|
| *Eif1* | 182 | **240** | **279** | 284 | 521 |
| *Kctd20* | 148 | **323** | **359** | 361 | 1405 |
| *Nabp1* | 485 | **663** | **705** | 725 | 1079 |
| *Nsd1* | 12 | **1076** | **1151** | 1171 | 8185 |
| *Snrpg* | 44 | **120** | **156** | 185 | 272 |

**B**

Canonical ORF

Truncated ORF

Overlapping iuORF

cTIS iuTIS ntTIS iuStop cStop

| | cTIS | iuTIS | ntTIS | iuStop | cStop |
|---|---|---|---|---|---|
| *Ccnt2* | 65 | **156** | 203 | **237** | 2234 |
| *Dhx16* | 135 | **2533** | 2586 | **2623** | 3267 |
| *Ep300* | 416 | **5787** | 5900 | **6066** | 7652 |
| *Etv5* | 133 | **683** | 739 | **749** | 1663 |
| *Hnrnpr* | 161 | **330** | 464 | **465** | 2057 |
| *Lamb1-1* | 129 | **218** | 273 | **473** | 5631 |
| *Mylpf* | 154 | **278** | 322 | **338** | 661 |
| *Nola2* | 46 | **192** | 280 | **309** | 505 |
| *Rbms2* | 154 | **24** | 226 | **317** | 1303 |
| *Rpf2* | 65 | **159** | 164 | **168** | 983 |
| *Sin3a* | 396 | **412** | 537 | **638** | 4218 |
| *Ssrp1* | 332 | **1914** | 2087 | **2163** | 2456 |
| *Tfdp1* | 120 | **454** | 480 | **481** | 1350 |
| *Trip12* | 126 | **1144** | 1191 | **1222** | 6201 |
| *Zfp281* | 170 | **222** | 290 | **459** | 2849 |

**Figure 7. mRNAs harboring iuORFs upstream of internal translation initiation sites.**

(A, B) Schematic representation of mRNAs harboring upstream (A) or overlapping (B) iuORFs. cTIS canonical translation initiation site, iuTIS and iuSTOP TIS and stop codon for the iuORF, ntTIS TIS for the N-terminally truncated proteoform ORF, cSTOP canonical stop codon. The numbers indicate the positions on the mRNA.

Finally, our work demonstrates that the translation of N-terminally truncated proteoforms can be regulated independently of the translation of the canonical ORF. Specifically, mTORC1 increased nTRIP6 translation without significantly affecting TRIP6 levels. Although unlikely, we cannot formally exclude the possibility that mTORC1 activity promotes the transcription of a *Trip6* mRNA variant that would code only for nTRIP6. The expression of the short KCTD20 proteoform, which is also repressed by an upstream iuORF was also increased upon transfection of the constitutively active RHEB mutant, strongly suggesting a stimulatory effect of mTORC1. mTORC1 has been shown to inhibit post-termination ribosome recycling after the translation of uORFs, and thereby to promote reinitiation at downstream start codons (Schepetilnikov et al, 2013; Gunišová et al, 2018; Zidek et al, 2015). Our results strongly suggest that mTORC1 stimulates nTRIP6 translation by a similar mechanism. Indeed, stimulation of mTORC1 activity by the constitutively active RHEB mutant, which increased nTRIP6 expression from the wild-type construct, did not increase it when the intercistronic distance was reduced to 5 nucleotides, a distance most likely too short for reinitiation. Similarly, mTORC1 did not stimulate nTRIP6 expression when the intercistronic distance was increased to 50 nucleotides, a distance most likely too long to repress reinitiation (Fig. 6A,B). Furthermore, mTORC1 inhibition prevented the increase in translation initiation at AUG2 which occurred during myoblast differentiation, while initiation at AUG1 and at the iuORF AUG was not affected. Thus, mTORC1 favors reinitiation at the nTRIP6 AUG after translation of the iuORF. Interestingly, the expression of the short TFDP1 proteoform was not increased upon transfection of the constitutively active RHEB mutant while that of the short KCTD20 proteoform was. The most likely explanation for this difference is that the iuORF in *Tfdp1* mRNA overlaps with the TIS for the short proteoform. Consequently, reinitiation downstream of the iuORF cannot occur at this TIS. Thus, given the similarities that we observed between the regulation of TRIP6 and KCTD20 short proteoforms, it is tempting to speculate that stimulation of reinitiation is a general mechanism by which mTORC1 increases the translation of the N-terminally truncated proteoforms after the translation of upstream iuORFs.

Our results reveal this mechanism operating in a physiological context. The transient increase in nTRIP6 expression at the transition between myoblast proliferation and differentiation (Norizadeh Abbariki et al, 2021) was inhibited by the PNA targeting AUG2, confirming that the increase is translational. Furthermore, the PNA treatment perfectly phenocopied the effect of blocking nTRIP6 function in myoblasts (Norizadeh Abbariki et al, 2021), which also confirms the specificity of the PNA effect, i.e., it accelerated early differentiation and inhibited late differentiation and fusion. Thus, the function of nTRIP6 in the temporal regulation of myoblast differentiation and fusion relies on a transient increase in translation at AUG2.

The transient increase in nTRIP6 expression during myoblast differentiation is preceded by a transient increase in mTORC1 activity and blocked upon mTORC1 inhibition. Thus, during early myogenesis, the increase in nTRIP6 expression occurs via an mTORC1-dependent stimulation of translation initiation at AUG2, downstream of the repressing iuORF. Interestingly, our results show that after peaking during late proliferation/early differentiation, mTORC1 activity starts to increase again at later time points.

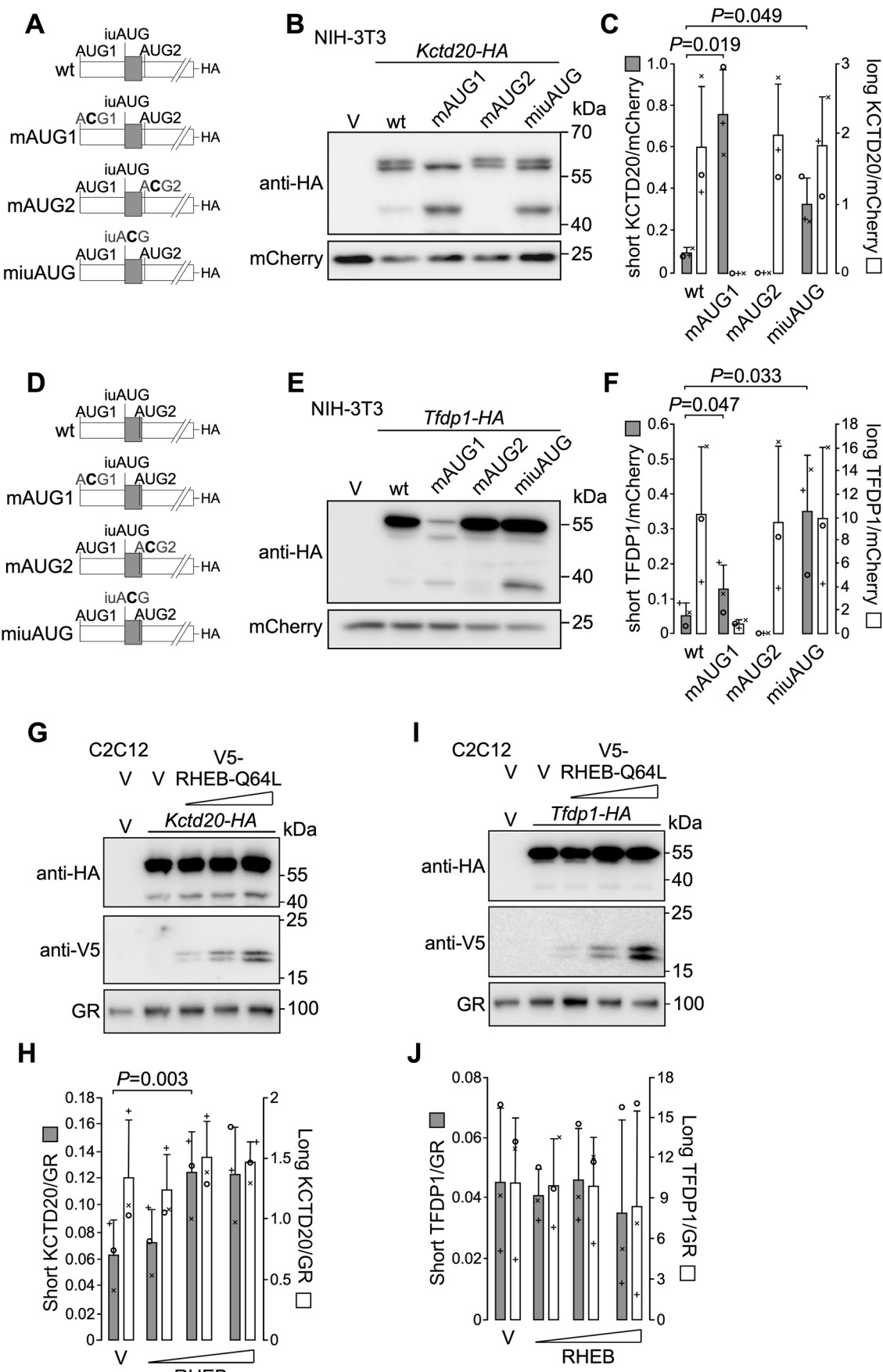

**Figure 8. iuORFs repress the translation of N-terminally truncated proteoforms.**

(**A**) Schematic representation of the *Kctd20* constructs used. The internal upstream open reading frame (iuORF) is indicated as a gray box. (**B, C**) NIH-3T3 fibroblasts were transfected with either an empty vector (V) or the indicated C-terminally tagged *Kctd20* construct together with mCherry as a transfection (TF) control. Cell lysates were analyzed by western blotting using anti-HA and anti-mCherry antibodies. Representative blots are shown (**B**). The expression of the short and long proteoforms of KCTD20 relative to the transfection control is presented as mean ± SD of three independent experiments (**C**). (**D**) Schematic representation *Tfdp1* constructs used. The internal upstream open reading frame (iuORF) is indicated as a gray box. (**E, F**) NIH-3T3 fibroblasts were transfected with either an empty vector (V) or the indicated C-terminally tagged *Tfdp1* construct together with mCherry as a transfection (TF) control. Cell lysates were analyzed by western blotting using anti-HA and anti-mCherry antibodies. Representative blots are shown (**E**). The expression of the short and long proteoforms of TFDP1 relative to the transfection control is presented as mean ± SD of three independent experiments (**F**). (**G–J**) Lysates of C2C12 cells transfected with either *Kctd20-HA* (**G, H**) or *Tfdp1-HA* (**I, J**) together with either an empty vector (V) or the V5-tagged constitutively active mutant RHEB-Q64L at increasing doses were subjected to western blotting using antibodies against the HA tag, the V5 tag and GR as a loading control. Representative western blots are shown (**G, I**). The expression of the short and of the long proteoforms of KCTD20 (**J**) or TFDP1 (**L**) relative to loading control is presented as mean ± SD of three independent experiments. In each graph, individual values are depicted by symbols, each representing an independent experiment. Student's paired *t* test *P* values are indicated. Source data are available online for this figure.

This observation is compatible with the reported stimulatory effect of mTORC1 on late differentiation and fusion (Erbay and Chen, 2001; Willett et al, 2009). However, other reports have shown that mTORC1 inhibits differentiation at early stages (Ge et al, 2011; Wilson et al, 2016). Given that nTRIP6 prevents premature differentiation (Norizadeh Abbariki et al, 2021) (Fig. 1), it is tempting to speculate that mTORC1 might inhibit early differentiation at least in part via the upregulation of nTRIP6 expression.

# Methods

## Reagents and tools table

| Reagent/resource | Reference or source | Identifier or catalog number |
|---|---|---|
| **Experimental models** | | |
| NIH-3T3 fibroblasts (mouse) | LGC Standards (ATCC) | Cat#ATCC-CRL-1658; RRID: CVCL_0594 |
| C2C12 myoblasts (mouse) | LGC Standards (ATCC) | Cat#ATCC-CRL-1772; RRID: CVCL_0188 |
| HEK-293 cells (human) | LGC Standards (ATCC) | Cat#ATCC-CRL-1573; RRID: CVCL_0045 |
| **Recombinant DNA** | | |
| Plasmid: pcDNA3.1-TRIP6-HA | This study | N/A |
| Plasmid: pcDNA3.1-TRIP6ΔAUG1-HA | This study | N/A |
| Plasmid: pcDNA3.1-TRIP6-mAUG2-HA | This study | N/A |
| Plasmid: pcDNA3.1-TRIP6-iuACG-HA | This study | N/A |
| Plasmid: pcDNA3.1-TRIP6ΔAUG1-iuACG-HA | This study | N/A |
| Plasmid: pcDNA3.1-TRIP6-V5 | This study | N/A |
| Plasmid: pcDNA3.1-TRIP6ΔAUG1-V5 | This study | N/A |
| pcDNA3.1-Hairpin-Trip6 HA-IRES-eGFP | This study | N/A |
| pcDNA3.1-control-Trip6 HA-IRES-eGFP | This study | N/A |
| pcDNA3.1-strong AUG1-TRIP6-HA | This study | N/A |

| Reagent/resource | Reference or source | Identifier or catalog number |
|---|---|---|
| pcDNA3.1-weak AUG1-TRIP6-HA | This study | N/A |
| pcDNA3.1-TRIP6-weak iuAUG-HA | This study | N/A |
| pcDNA3.1-weak AUG1-TRIP6-weak iuAUG -HA | This study | N/A |
| pcDNA3.1-TRIP6-iuORF-5nt-AUG2-HA | This study | N/A |
| pcDNA3.1-TRIP6-iuORF-50nt-AUG2-HA | This study | N/A |
| pcDNA3.1-PNA-target-Luciferase | This study | N/A |
| pcDNA3.1-misPNA-target-Luciferase | This study | N/A |
| pcDNA3.1-Renilla | This study | N/A |
| Plasmid: pcDNA3.1-V5-RHEB-Q64L | This study | N/A |
| Plasmid: pcDNA3.1-KCTD20-HA | This study | N/A |
| Plasmid: pcDNA3.1-KCTD20-mAUG1-HA | This study | N/A |
| Plasmid: pcDNA3.1-KCTD20-mAUG2-HA | This study | N/A |
| Plasmid: pcDNA3.1-KCTD20-miuAUG-HA | This study | N/A |
| Plasmid: pcDNA3.1-TFDP1-HA | This study | N/A |
| Plasmid: pcDNA3.1-TFDP1-mAUG1-HA | This study | N/A |
| Plasmid: pcDNA3.1-TFDP1-mAUG2-HA | This study | N/A |
| Plasmid: pcDNA3.1-TFDP1-miuAUG-HA | This study | N/A |
| Plasmid: pcDNA3.1-eGFP | This study | N/A |
| Plasmid: pcDNA3.1-mCherry | This study | N/A |
| Plasmid library: Medaka kinases cloned into pCMV-Sport6.1 | Chen et al, 2014; Souren et al, 2009 | N/A |
| **Antibodies** | | |

| Reagent/resource | Reference or source | Identifier or catalog number |
|---|---|---|
| Rabbit monoclonal anti-TRIP6 | Kemler et al, 2016 | N/A |
| Mouse monoclonal anti-Glucocorticoid Receptor (GR) | Santa Cruz Biotechnology | Cat#sc-393232; RRID: AB_2687823 |
| Mouse monoclonal anti-V5 tag | Thermo Fisher Scientific | Cat#R960-25; RRID: AB_2556564 |
| Rat monoclonal anti-HA tag (clone 3F10) | Roche | Cat#11867431001; RRID: AB_390919 |
| Rabbit monoclonal anti-GFP (clone D5.1) | Cell Signaling Technology | Cat#2956; RRID: AB_1196615 |
| Rabbit monoclonal anti-phospho (Thr389)-p70 S6 Kinase (clone 108D2) | Cell Signaling Technology | Cat#9234; RRID: AB_2269803 |
| Rabbit monoclonal anti-p70 S6 Kinase (clone 49D7) | Cell Signaling Technology | Cat#2708; RRID: AB_390722 |
| Rabbit polyclonal anti-mCherry | Abcam | Cat#ab167453; RRID: AB_2571870 |
| Mouse monoclonal anti-MYH3 (embryonic myosin heavy chain) | DSHB | Cat#F1.652; RRID: AB_528358 |
| **Oligonucleotides and other sequence-based reagents** | | |
| Cell-penetrating peptide nucleic acids (PNAs) | This study | Appendix Table S1 |
| Primers used for the identification of alternative Trip6 mRNAs | This study | Appendix Table S2 |
| Real-time PCR primers | Norizadeh Abbariki et al, 2021 and this study | Appendix Table S3 |
| Oligonucleotides used in the translation initiation assay | This study | Appendix Table S4 |
| **Chemicals, enzymes, and other reagents** | | |
| Harringtonine | Biomol | Cat# Cay15361; CAS: 26833-85-2 |
| Cycloheximide | ThermoFisher Scientific | Cat# J66901.03; CAS: 66-81-9 |
| RNase I | ThermoFisher Scientific | Cat# AM2295 |
| SUPERase·In RNase inhibitor | ThermoFisher Scientific | Cat# AM2696 |
| T4 polynucleotide kinase | Promega | Cat# M4103 |
| polyA polymerase | ThermoFisher Scientific | Cat# AM2030 |
| DAPI | Thermo Fisher Scientific | Cat# 62248; CAS: 28718-90-3 |
| Everolimus | Sigma-Aldrich | Cat# SML2282; CAS: 159351-69-6 |
| **Software** | | |
| Fiji | Schindelin et al, 2012 | https://imagej.net/software/fiji/ |
| R | | https://www.r-project.org |
| Image Lab | Bio-Rad laboratories | https://www.bio-rad.com/de-de/product/image-lab-software |

| Reagent/resource | Reference or source | Identifier or catalog number |
|---|---|---|
| EMBOSS version 6.5.7 | Rice et al, 2000 | https://emboss.sourceforge.net |
| **Other** | | |
| T7 TNT Coupled Reticulocyte Lysate System | Promega | Cat# L4611 |
| QuikChange site-directed mutagenesis kit | Agilent | Cat# 200513 |
| Zero Blunt TOPO PCR cloning kit | ThermoFisher Scientific | Cat# 450245 |
| FirstChoice RLM-RACE Kit | ThermoFisher Scientific | Cat# AM1700M |

## Methods and protocols

### Plasmid constructs

In pcDNA3.1-TRIP6-HA the mouse *Trip6* CDS with a C-terminal HA epitope was cloned by PCR between the NheI and EcoRI sites of pcDNA3.1 (Invitrogen, Karlsruhe, Germany). The KpnI/XhoI fragment of pcDNA3.1-TRIP6-HA, lacking the first ATG, was cloned into pcDNA3.1 to obtain pcDNA3.1-TRIP6ΔAUG1-HA. pcDNA3.1-TRIP6-mAUG2-HA and pcDNA3.1-TRIP6-iuACG-HA were obtained by mutating the nTRIP6 ATG (ATG2) and the iuORF ATG (iuATG) into ACG, respectively, using the Quik-Change site-directed mutagenesis kit (Agilent, Waldbronn, Germany). The KpnI/XhoI fragment of pcDNA3.1-TRIP6-iuACG-HA was cloned into pcDNA3.1 to obtain pcDNA3.1-TRIP6ΔAUG1-iuACG-HA. The *Trip6* CDS constructs C-terminally tagged with the V5 epitope pcDNA3.1-TRIP6-V5 and pcDNA3.1-TRIP6-ΔAUG1-V5 were cloned using a similar strategy. pcDNA3.1-Hairpin-Trip6 HA-IRES-eGFP was cloned as follows: a single AgeI site was introduced by oligonucleotide cloning immediately after the pcDNA3.1 transcription start site. Subsequently, an oligonucleotide containing the hairpin sequence (GTCCACCACGGCC-GATATCACGGCCGTGGTGGAC) was cloned between the SacI and AgeI sites of the resulting vector. The TRIP6-HA coding sequence was cloned between the NheI and EcoRI sites and the EMCV IRES sequence driving eGFP translation was cloned as a string DNA fragment between the EcoRI and XhoI sites. pcDNA3.1-control-Trip6 HA-IRES-eGFP containing the control sequence without secondary structure (CAACAACAACAACAA-CAACAACAACAACAACAAC) was cloned using the same strategy. pcDNA3.1-strong AUG1-TRIP6-HA and pcDNA3.1-weak AUG1-TRIP6-HA were obtained by PCR amplifying the *Trip6* coding sequence using the following forward primers containing a 5'- NheI site, GCTGGCTAGCGCCACCATGGCCGGGCC-CACCTG and AACTCGCTAGCCTTTTTATGTCCGGGCC-CACCTG, respectively, and a reverse primer within *Trip6* coding sequence encompassing a KpnI site. The PCR products were then subcloned into pcDNA3.1-TRIP6-HA. pcDNA3.1-TRIP6-weak iuAUG-HA was obtained by introducing a double-stranded oligonucleotide (GGGGGGCCTACTTGGGTGGctttttATGtaA-CACCCcagcgcctgca) into pcDNA3.1-TRIP6-HA linearized with BamHI using In-Fusion cloning (Takara Bio Europe, Saint-Germain-en-Laye, France). pcDNA3.1-weak AUG1-TRIP6-weak

iuAUG -HA was obtained by subcloning the NheI/KpnI fragment of pcDNA3.1-weak AUG1-TRIP6-HA into pcDNA3.1-TRIP6-weak iuAUG-HA. pcDNA3.1-TRIP6-iuORF-5nt-AUG2-HA and pcDNA3.1-TRIP6-iuORF-50nt-AUG2-HA, in which the intercistronic distance between the iuORF stop and AUG2 has been decreased to 5 nucleotides and increased to 50 nucleotides, respectively, were obtained by subcloning the respective string DNA fragment between the BamHI and SacII sites of pcDNA3.1-TRIP6-HA. The reporter construct to assess the effectiveness of the PNA targeting AUG2 in *Trip6* mRNA, pcDNA3.1-PNA-target-Luciferase was cloned as follows: the firefly luciferase coding sequence from pGL3-basic (Promega, Mannheim, Germany) was subcloned into pcDNA3.1; the PNA target sequence (ATGTTGGCTGATCTGGAC) was then introduced in-frame with luciferase as a string DNA fragment, which removed the luciferase translation initiation site. The control construct to assess the specificity of the PNA, pcDNA3.1-misPNA-target-Luciferase was cloned using the same strategy with the PNA target containing 3 mismatches (ATGTTAGCTTATGTGTAC). pcDNA3.1-Renilla was obtained by subcloning the *Renilla reniformis* luciferase coding sequence from pUbi-Renilla (Kassel et al, 2004) into pcDNA3.1. The constitutively active RHEB mutant RHEB-Q64L (Jiang and Vogt, 2008) N-terminally tagged with the V5 epitope was synthesized as a DNA string (GeneArt Strings DNA Fragments, Thermo Fisher Scientific, Schwerte, Germany) and cloned into pcDNA3.1 to obtain pcDNA3.1-V5-RHEB-Q64L. C-terminally HA-tagged KCTD20 and TFDP1 constructs were cloned as synthesized DNA strings into pcDNA3.1. The QuikChange site-directed mutagenesis kit was used to mutate the canonical initiation codon (AUG1), the initiation codon for the N-terminally truncated proteoform (AUG2) and the initiation codon for the short internal ORF (iuAUG) into non-initiating codons (ACG) to obtain pcDNA3.1-KCTD20-mAUG1-HA, pcDNA3.1-KCTD20-mAUG2-HA, pcDNA3.1-KCTD20-miuAUG-HA, pcDNA3.1-TFDP1-mAUG1-HA, pcDNA3.1-TFDP1-mAUG2-HA and pcDNA3.1-TFDP1-miuAUG-HA. pcDNA3.1-eGFP and pcDNA3.1-mCherry were obtained by cloning the corresponding string DNA fragments (ThermoFisher Scientific) into pcDNA3.1. All constructs were verified by sequencing and are available upon reasonable request.

### Cell culture, transfection, and cellular assays

C2C12 myoblasts, NIH-3T3 fibroblasts, and HEK-293 cells (all from ATCC, LGC Standards GmbH, Wesel, Germany) were cultured in Dulbecco's modified Eagle's medium (DMEM) supplemented with 10% fetal calf serum (FCS). Cell lines were authenticated by morphological examination and were routinely checked for mycoplasma contamination.

We used a standardized protocol for the differentiation of C2C12 myoblasts (Norizadeh Abbariki et al, 2021). Cells were seeded at a density of $5 \times 10^3$ cells/cm$^2$ in growth medium (GM, 10% FCS-containing DMEM) at day $-3$, relative to the induction of differentiation at day 0. When cells reached confluence at day 0, differentiation was induced by changing the medium to differentiation medium (DM, 2% horse serum-containing DMEM), which was then replaced every second day. Where indicated, cells were treated with the mTORC1 inhibitor Everolimus (Sigma-Aldrich) at a concentration of 100 nM. NIH-3T3 fibroblasts and C2C12 myoblasts were transfected using *Trans*IT-X2 (Mirus Bio LLC) and HEK-293 cells with ScreenFect-A (ScreenFect,

Eggenstein-Leopoldshafen, Germany). A PNA was designed to target the mouse *Trip6* mRNA sequence (NM_011639.3) from positions 481 to 498 which encompass AUG2. A mispaired PNA (Misp) and in some experiments, an unrelated, random PNA sequence were used as controls (see sequences in Appendix Table S1). The cell-penetrating moiety of the PNAs consists of an octoarginine peptide (R8). The PNAs were synthesized manually by standard Fmoc-based PNA solid-phase protocols as described (Vázquez and Seitz, 2014) (TentaGel® S RAM resin; 10 μmol scale; after the 12$^{th}$ PNA-monomer coupling, the used equivalents increased from 4 to 8 for 1 h). Peptide-PNA conjugates were purified by preparative reverse-phase (RP) HPLC and identified by high-resolution mass spectrometry (Appendix Fig. S2). To increase the penetration, the PNAs were applied to C2C12 cells at a concentration of 10 μM in DMEM with reduced FCS (2.5%) for 2 h, after which FCS concentration was increased to 10%.

For luciferase activity measurements, NIH-3T3 fibroblasts were transfected with the PNA luciferase reporter genes together with pcDNA3.1-Renilla to normalize the activity for possible differences in transfection efficiency. Cells were treated 4 h later with the cell-penetrating PNAs and harvested 24 h later for the determination of luciferase activity as described previously (Diefenbacher et al, 2014). Firefly luciferase activities were normalized to Renilla luciferase activities within the same samples.

For the kinase overexpression screen, a library of 184 unique Medaka kinases cloned into pCMV-Sport6.1 (Chen et al, 2014; Souren et al, 2009) was kindly provided by Jochen Wittbrodt (Centre for Organismal Studies, Heidelberg, Germany) and Gary Davidson (Institute for Biological and Chemical Systems, Karlsruhe Institute of Technology, Karlsruhe, Germany). HEK-293 cells were co-transfected in 96-well plates with the V5-tagged TRIP6 construct and either an empty vector as a control or the library using ScreenFect-A. Cells were lysed 24 h later, and the relative expression of TRIP6 and nTRIP6 was assessed by western blot analysis (see below).

### Identification of alternative *Trip6* mRNAs

In order to identify putative splice variants that could encode for nTRIP6, total RNA was extracted from NIH-3T3 cells using PeqGOLD TriFast (Peqlab Biotechnologie, Erlangen, Germany) and reverse-transcribed using M-MLV Reverse Transcriptase, RNase H Minus and random hexamers (Promega). The resulting cDNA was amplified by PCR using a forward primer located in the 5'-untranslated region and a reverse primer located in exon 9 of *Trip6* mRNA (see primer sequences in Appendix Table S2). The resulting amplicons were subcloned using the Zero Blunt TOPO cloning kit (ThermoFisher Scientific) and sequenced. Identification of alternative *Trip6* transcript from putative alternative promoters was performed by RNA Ligase Mediated Rapid Amplification of cDNA Ends (RLM-RACE) using the FirstChoice RLM-RACE Kit (ThermoFisher Scientific) according to the manufacturer's instructions, starting with total RNA extracted from NIH-3T3 cells. The sequences of the *Trip6*-specific outer and inner reverse primers are described in Appendix Table S2. The resulting amplicons were subcloned using the Zero Blunt TOPO cloning kit and sequenced.

### RNA isolation and quantitative real-time PCR (qRT-PCR)

Total RNA was extracted using PeqGOLD TriFast and reverse-transcribed into cDNA. *Trip6*, *Myog* (myogenin) and *Tnni2* mRNAs, as well as the transcript of the large ribosomal subunit

P0 gene (*Rplp0*) used for normalization, were quantified by real-time PCR using the ABI Prism Sequence Detection System 7000 (Applied Biosystems, Foster City, CA). The primers (ThermoFisher Scientific) are described in Appendix Table S3.

## Western blotting

Western blotting analyses were performed using the following antibodies: a custom-made rabbit anti-TRIP6 monoclonal antibody (Kemler et al, 2016); anti-V5 (R960-25, Thermo Fisher Scientific); anti-HA (clone 3F10, Roche Applied Science, Mannheim, Germany); anti-GFP (clone D5.1, Cell Signaling Technology); anti-mCherry (Ab167453, Abcam); anti-phospho (Thr389)-p70 S6 Kinase (clone 108D2, Cell Signaling Technology). To normalize the phospho (Thr389)-p70 S6 Kinase signals membranes were stripped and re-probed with a phosphorylation-independent anti-p70 S6 Kinase antibody (clone 49D7, Cell Signaling Technology). The expression of the "classical" housekeeping genes β-actin and GAPDH is regulated during myogenesis (Hildyard and Wells, 2014; Otey et al, 1988; Cox et al, 1990). Therefore, we selected the glucocorticoid receptor (GR) as a loading control, given that it is stably expressed during C2C12 differentiation (Sun et al, 2008). To this end, we used an anti-GR antibody (sc-393232, Santa Cruz, Heidelberg, Germany). Signals were detected by enhanced chemoluminescence using the ChemiDoc Touch Imaging System (Bio-Rad Laboratories, Munich, Germany). Signal quantification was performed within the linear range of detection using the Image Lab software (Bio-Rad laboratories). Linear brightness and contrast adjustments were made for illustration purposes only after the analysis had been made.

## In vitro transcription and translation

The plasmids pcDNA3.1-TRIP6-V5 and pcDNA3.1-TRIP6-ΔAUG1-V5, as well as pcDNA3.1-eGFP as a control were used as templates for in vitro transcription and translation using the T7 TNT® Coupled Reticulocyte Lysate System (Promega) according to the manufacturer's instructions, except that the reactions were performed for 90 min at 30 °C in the presence of 5 μM PNA targeting the 2nd AUG of *Trip6* mRNA or its mispaired control. The translated products were analyzed by western blotting.

## Translation initiation assay

A ribosome profiling protocol (Ingolia et al, 2012b) was adapted in order to assess translation initiation at the various AUGs of *Trip6* mRNA. C2C12 myoblasts were treated at day −2 or at day 0 of a differentiation experiment with 2 μg/ml Harringtonine (Biomol) for 2 min at 37 °C to arrests 80 s ribosomes immediately after initiation (Ingolia et al, 2011b, 2012). Cells were washed twice in ice-cold PBS, harvested in ice-cold lysis buffer (20 mM Tris·Cl pH 7.4, 150 mM NaCl, 1.5 mM MgCl2, 1 mM DTT, 100 μg/ml cycloheximide, 1% Triton X-100). The lysates were treated with 2.5 U/μl RNase I (Life Technologies) and the nuclease digestion was stopped by addition of 1 U/μl of SUPERase·In RNase inhibitor (ThermoFisher Scientific). Ribosomes were pelleted through a sucrose cushion (1 M sucrose, 20 mM Tris·Cl pH 7.4, 150 mM NaCl, 1.5 mM MgCl2, 1 mM DTT, 100 μg/ml cycloheximide, 20 U/ml SUPERase·In) by centrifugation at 70,000 r.p.m. at 4 °C for 4 h. The ribosome-protected RNA fragments were then extracted using PeqGOLD TriFast, dephosphorylated using T4 polynucleotide kinase (Promega), polyA tailed using polyA polymerase

(ThermoFisher Scientific) and reverse-transcribed using an oligo dT adapter. Fragments containing the *Trip6* mRNA translation initiation sites (AUG1, AUG2, and uAUG), a *Trip6* mRNA non-initiating fragment as a negative control, the GR mRNA translation initiation site used for normalization as well as a GR mRNA non-initiating fragment as a negative control were detected and quantified by real-time PCR using a sequence-specific forward primer and a universal reverse primer corresponding to the adapter used for reverse transcription. The sequences of the oligonucleotides are provided in Appendix Table S4. An initiation index at each AUG in *Trip6* mRNA was calculated relative to the initiation of GR mRNA translation after correction for the background signal of the negative controls (*Trip6* and GR non-initiating fragments) by the ΔΔCt method.

## In silico identification of iuORFs

In order to identify other mRNAs harboring short internal ORFs that putatively regulate the translation of N-terminally truncated proteoforms, we curated a published ribosome profiling dataset (Ingolia et al, 2011b; Data ref: Ingolia et al, 2011a). Out of the 4994 mRNAs analyzed, we first selected those in which the canonical TIS was identified and that harbored at least one internal (3' to the canonical TIS) out-of-frame, short (≤ 100 codons) ORF and at least one in-frame internal TIS for a truncated proteoform. The resulting mRNAs were then filtered to keep only those in which the TIS of the short ORF was located 5' to the truncated proteoform TIS, and in which the stop codon of the short ORF was located either 3' to the truncated proteoform TIS (overlapping short ORF) or at a maximum distance of 30 nucleotide 5' to the truncated proteoform TIS (upstream short ORF). When several short ORFs fulfilled these criteria within the same mRNA, only the one with the highest initiation score (Ingolia et al, 2011b) was kept. We then performed a conservation analysis of the features most relevant for the translation of N-terminally truncated proteoforms and for short ORFs to regulate initiation at downstream AUGs (Chen and Tarn, 2019; Zhang et al, 2019), i.e., the sequence contexts of the canonical, short ORF and truncated proteoform initiation codons relative to the Kozak sequence (Kozak, 2002), the distance from the canonical TIS to the short ORF TIS, the distance from the short ORF stop codon to the truncated proteoform TIS and the length of the short ORF. Human and rat orthologues to mouse genes were selected using the closest homolog in the phylogenetic tree using clustalw. Conservation of the start codons and length of the ORF were obtained by screening the multiple alignments with the EMBOSS (Rice et al, 2000) functions extractalign and transeq.

## Immunofluorescence, microscopy, and image analysis

Immunofluorescence analysis was performed on C2C12 cells grown and differentiated on glass coverslips coated with collagen Type I (Sigma-Aldrich), fixed for 10 min in 10% formalin, permeabilized for 10 min in 0.5% Triton X-100 in PBS and blocked for 1 h in 5% BSA in PBS. The primary antibody was a mouse anti-MYH3 (F1.652-b, Developmental Studies Hybridoma Bank, deposited by Blau HM) and the secondary antibody was an Alexa Fluor 488-conjugated anti-mouse antibody (Invitrogen). Nuclei were counter-stained with DAPI (Thermo Fisher Scientific). Cells were imaged by confocal microscopy on a Zeiss LSM 800 (Zeiss, Jena, Germany). Cells images were acquired in tiling mode using a 10×/0.3 Plan-Neofluar objective resulting in 3 × 2 mm² images. Images were analyzed using Fiji (Schindelin et al, 2012). The number of MYH3-

positive mononuclear cells was determined by a combination of automated segmentation and manual counting. The fusion index was calculated as the percentage of nuclei within fused myotubes. Linear brightness and contrast adjustments were made for illustration purposes, but only after the analysis had been made.

### Statistical analysis

Statistical analyses were performed using R. Where indicated, significant differences were assessed by two-sided $t$ test analysis, with values of $P < 0.05$ sufficient to reject the null hypothesis. A false discovery rate (FDR) correction (Benjamini and Hochberg, 1995) was applied when multiple comparisons were performed. For the kinase overexpression screen, the normal distribution of the nTRIP6/TRIP6 ratio was first confirmed by a Shapiro–Wilk normality test ($W = 0.9803$, $P = 1.07 \times 10^{-2}$) as well as by the Kernel density and Q-Q plots (Fig. EV2). This allowed us to calculate Z scores and to derive two-sided $P$ values in order to determine the "hits", kinases which significantly altered the nTRIP6/TRIP6 ratio.

## Data availability

Source data for Fig. 3F have been deposited at BioImage Archive (https://www.ebi.ac.uk/biostudies/bioimages/studies/S-BIAD1566).

The source data of this paper are collected in the following database record: biostudies:S-SCDT-10_1038-S44319-025-00390-z.

## Peer review information

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

## Acknowledgements

The authors are grateful to Jochen Wittbrodt (Centre for Organismal Studies, Heidelberg, Germany) and Gary Davidson (Institute for Biological and Chemical Systems, Karlsruhe Institute of Technology, Karlsruhe, Germany) for providing the Medaka kinase library. The authors thank Nicholas S Foulkes and Andrew CB Cato for stimulating discussions and critically reading the manuscript. This work was supported in part by the German Science Foundation (grants 315384510 to OK and 425970020 to OK and OV). The authors acknowledge support by the KIT-Publication Fund of the Karlsruhe Institute of Technology.

## Author contributions

**Raphael Fettig**: Investigation; Methodology. **Zita Gonda**: Investigation; Methodology. **Niklas Walter**: Investigation. **Paul Sallmann**: Investigation. **Christiane Thanisch**: Investigation; Methodology. **Markus Winter**: Investigation; Methodology. **Susanne Bauer**: Investigation. **Lei Zhang**: Resources. **Greta Linden**: Resources. **Margarethe Litfin**: Resources; Investigation; Methodology. **Marina Khamanaeva**: Investigation. **Sarah Storm**: Investigation. **Christina Münzing**: Investigation. **Christelle Etard**: Resources; Investigation; Writing—review and editing. **Olivier Armant**: Methodology; Writing—review and editing. **Olalla Vázquez**: Conceptualization; Supervision; Funding acquisition; Methodology; Writing—review and editing. **Olivier Kassel**: Conceptualization; Supervision; Funding acquisition; Methodology; Writing—original draft; Writing—review and editing.

Source data underlying figure panels in this paper may have individual authorship assigned. Where available, figure panel/source data authorship is listed in the following database record: biostudies:S-SCDT-10_1038-S44319-025-00390-z.

## Funding

## Disclosure and competing interests statement

The authors declare no competing interests.

# Expanded View Figures

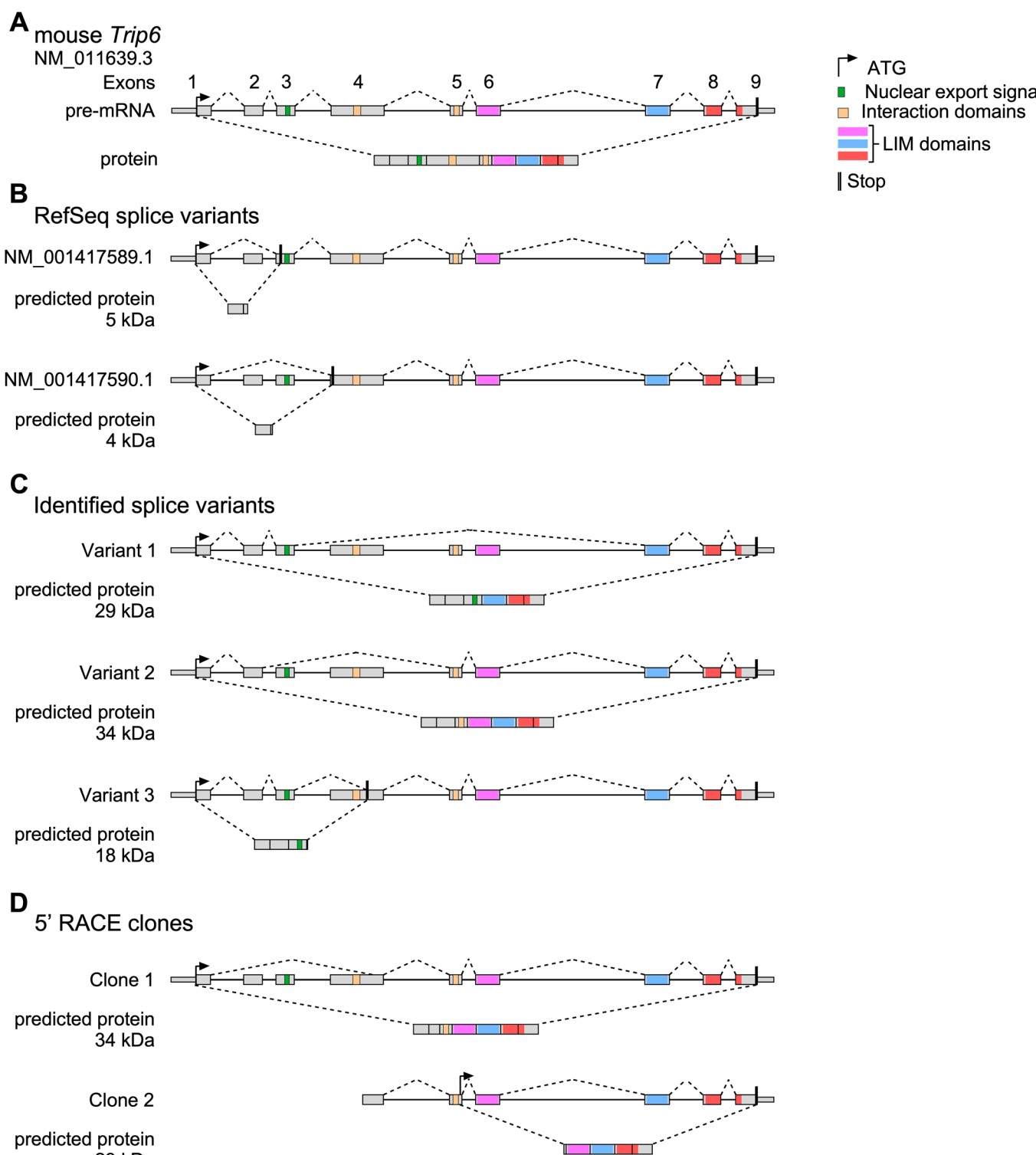

**Figure EV1. Schematic representation of the identified *Trip6* mRNA variants.**

(A) Exon/intron structure of mouse *Trip6* pre-mRNA and functional domain of TRIP6 protein. (B–D) Schematic representation of the published (B) or identified (C, D) *Trip6* pre-mRNA variants with the predicted translation products domain structure and molecular weight.

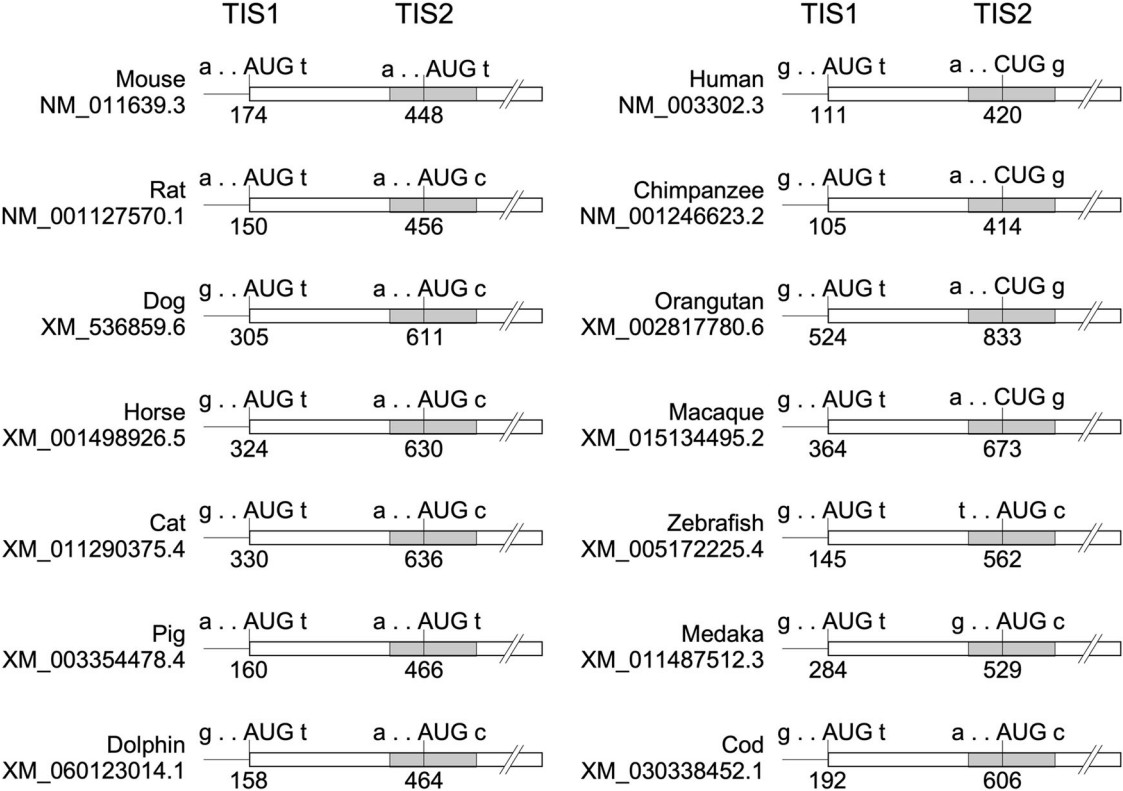

**Figure EV2. Conservation of TRIP6 and nTRIP6 translation initiation sites in *Trip6* mRNA.**

For each species, the schematic representation of the *Trip6* mRNA depicts the sequences surrounding the translation initiation site of TRIP6 (TIS1) and of nTRIP6 (TIS2) relative to the Kozak sequence (RNN AUG GNN where N represents any nucleotide and R a purine). The numbers indicate the positions on the mRNA. The gray box represents the Nuclear Export Signal encoding sequence.

**A**

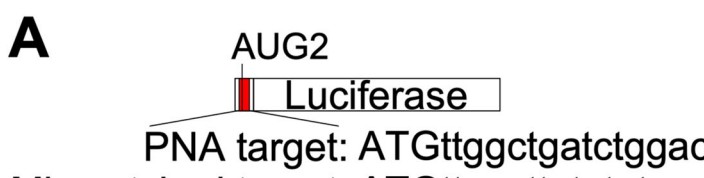

AUG2

Luciferase

PNA target: ATGttggctgatctggac

Mismatched target: ATGtt<u>a</u>gctt<u>a</u>t<u>g</u>t<u>g</u>tac

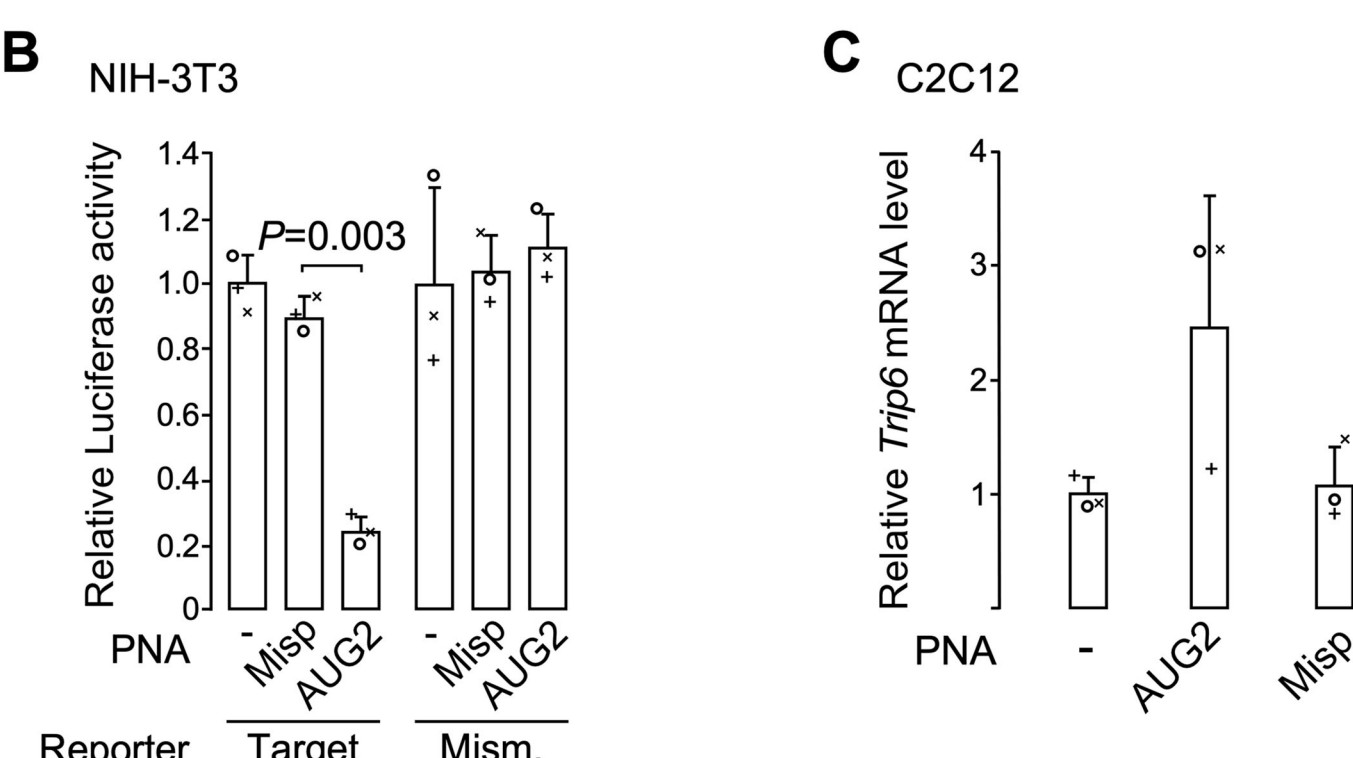

**B** NIH-3T3

**C** C2C12

Figure EV3. Validation of the AUG2-targeting PNA.

(A) Schematic representation of the reporter constructs used. The PNA target sequence encompassing AUG2 (uppercase ATG) or a mismatched sequence (underlined mismatches) was fused in-frame with Firefly luciferase. (B) NIH-3T3 fibroblasts were transfected with the target or mismatched (Mism.) reporter construct together with an expression vector for Renilla luciferase and 4 h later mock-treated (-) or treated with the mispaired control PNA (Misp.) or the AUG2-targeting PNA. Normalized luciferase activities were determined 24 h later and are plotted relative to the target reporter transfected, mock-treated cells (mean ± SD of three independent experiments). (C) C2C12 myoblasts were mock-treated or treated with the indicated PNA for 24 h. Relative levels of *Trip6* mRNA were determined by reverse transcription and real-time PCR. Results are plotted relative to the expression of the *Rplp0* gene (mean ± SD of three independent experiments). In each graph individual values are depicted by symbols, each representing an independent experiment. Student's paired *t* test *P* values are indicated. Source data are available online for this figure.

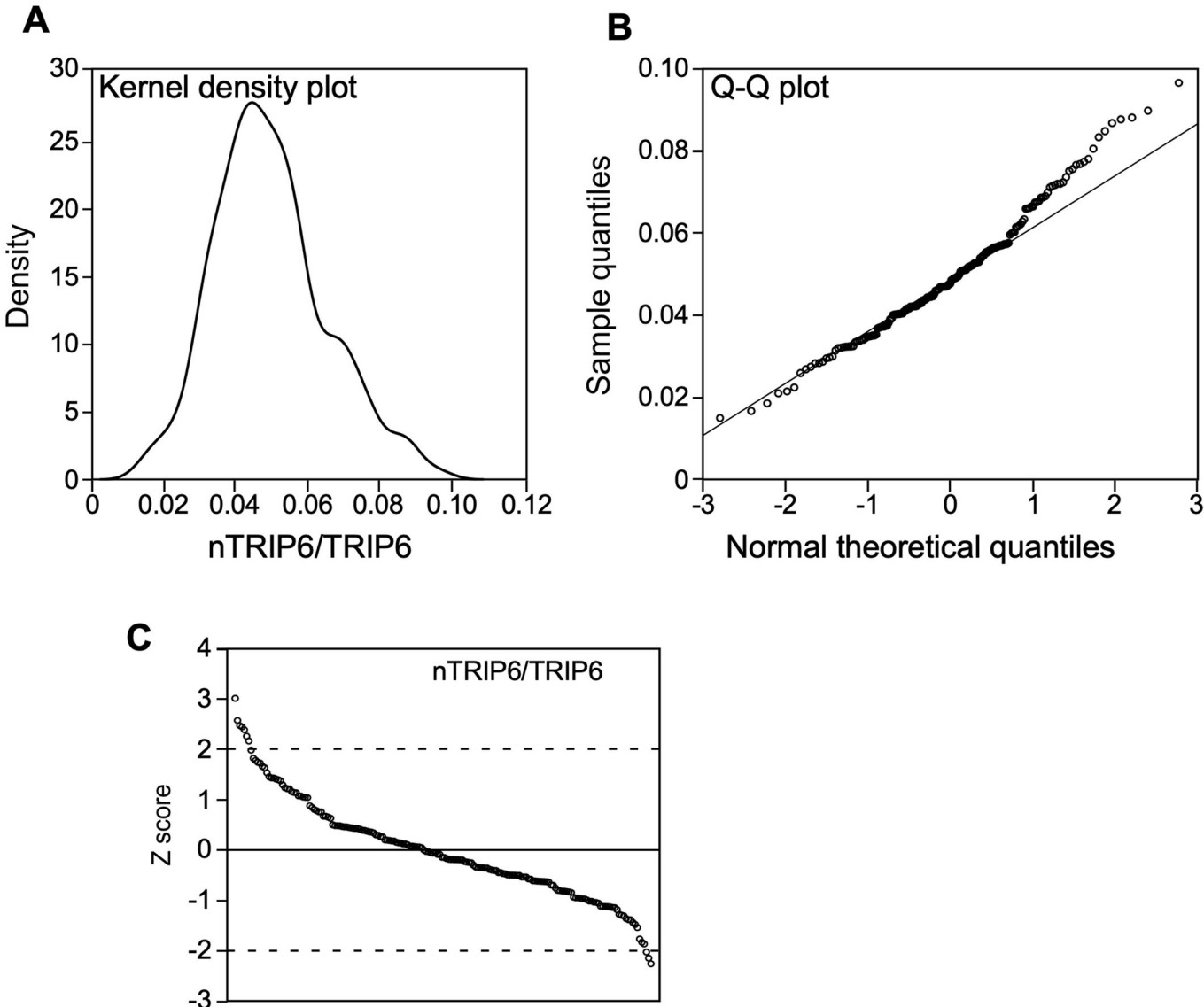

**Figure EV4. Relative expression of nTRIP6 and TRIP6 in the kinase overexpression screen.**

The nTRIP6/TRIP6 ratios were determined by western blotting analysis in HEK-293 co-transfected with the C-terminally V5-tagged Trip6 CDS construct and a library of 184 unique Medaka kinases. (**A**, **B**) The ratios are normally distributed. The Kernel density and Q-Q plots are presented. Shapiro–Wilk normality test: $W = 0.9803$, $P = 1.07 \times 10\text{-}2$. (**C**) Z scores of the nTRIP6/TRIP6 ratios.

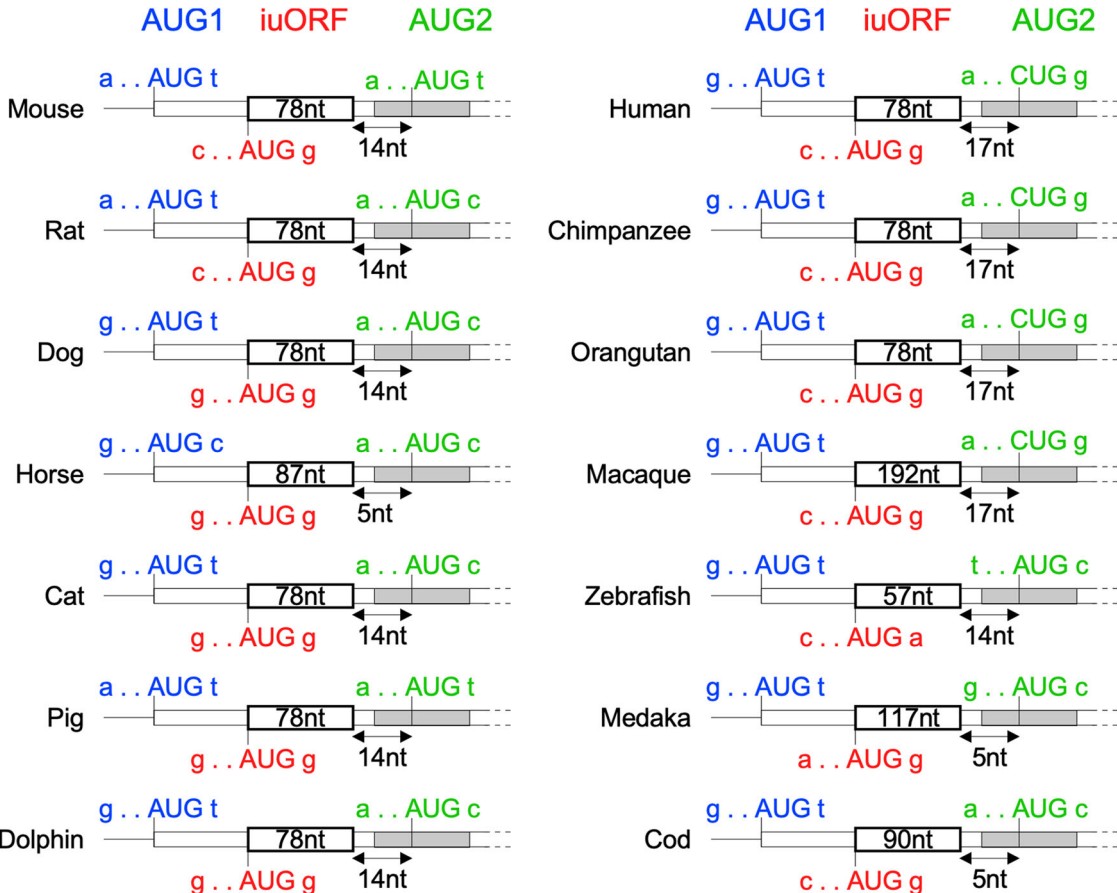

**Figure EV5. Conservation of the internal upstream open reading frame (iuORF) in Trip6 mRNA.**

For each species, the schematic representation of the 5' of Trip6 mRNA depicts the surrounding sequence of the TRIP6 (AUG1), iuORF and nTRIP6 (AUG2) initiation codons relative to the Kozak sequence (RNN AUG GNN where N represents any nucleotide and R a purine), the number of nucleotides (nt) in the uORF and the distance between the iuORF and AUG2. The gray box represents the Nuclear Export Signal encoding sequence.

