## [Peer Review File · EMBO Reports]

Short internal open reading frames repress the translation of N-terminally truncated proteoforms

Raphael Fettig, Zita Gonda, Niklas Walter, Paul Sallmann, Christiane Thanisch, Markus Winter, Susanne Bauer, Lei Zhang, Greta Linden, Margarethe Litfin, Marina Khamanaeva, Sarah Storm, Christina Münzing, Christelle Etard, Olivier Armant, Olalla Vázquez, and Olivier Kassel

Corresponding author(s): Olivier Kassel (olivier.kassel@kit.edu)

Review Timeline:

Submission Date:	13th Feb 24
Editorial Decision:	22nd Apr 24
Revision Received:	9th Jan 25
Editorial Decision:	29th Jan 25
Revision Received:	31st Jan 25
Accepted:	4th Feb 25

Editor: Esther Schnapp

Transaction Report:

Dear Olivier,

Thank you for the submission of your manuscript to EMBO reports. We have now received the full set of referee reports that is pasted below.

As you will see, the referees acknowledge that the findings are potentially interesting. However, referees 2 and 3 rate the technical quality of the study "low/unacceptable" in the ms summary table that is directly sent to the editor, and all referees raise concerns and have several suggestions for how the study should be improved. Given that you think that you can address all concerns, as we discussed, I would like to invite you to revise your manuscript with the understanding that the referee concerns must be fully addressed and their suggestions taken on board.

Please address all referee concerns in a complete point-by-point response. Acceptance of the manuscript will depend on a positive outcome of a second round of review. It is EMBO reports policy to allow a single round of major revision only and acceptance or rejection of the manuscript will therefore depend on the completeness of your responses included in the next, final version of the manuscript.

We realize that it is difficult to revise to a specific deadline. In the interest of protecting the conceptual advance provided by the work, we recommend a revision within 3 months (23rd Jul 2024). Please discuss the revision progress ahead of this time with the editor if you require more time to complete the revisions.

- 1) A data availability section providing access to data deposited in public databases is missing. If you have not deposited any data, please add a sentence to the data availability section that explains that.
- 2) Your manuscript contains statistics and error bars based on $n=2$. Please use scatter blots in these cases. No statistics should be calculated if $n=2$.

3) We replaced Supplementary Information with Expanded View (EV) Figures and Tables that are collapsible/expandable online. A maximum of 5 EV Figures can be typeset. EV Figures should be cited as 'Figure EV1, Figure EV2' etc... in the text and their respective legends should be included in the main text after the legends of regular figures.

5) a complete author checklist, which you can download from our author guidelines . Please insert information in the checklist that is also reflected in the manuscript. The completed author checklist will also be part of the RPF.

6) Please note that all corresponding authors are required to supply an ORCID ID for their name upon submission of a revised manuscript (). Please find instructions on how to link your ORCID ID to your account in our manuscript tracking system in our Author guidelines

- the name of the statistical test used to generate error bars and P values,
- the number (n) of independent experiments (please specify technical or biological replicates) underlying each data point,
- the nature of the bars and error bars (s.d., s.e.m.),
- If the data are obtained from n {less than or equal to} 2, use scatter blots showing the individual data points.

I look forward to seeing a revised form of your manuscript when it is ready.

Best regards,

Esther

Referee #1:

The manuscript describes the translational regulation of TRIP6 and an N-terminally truncated protein isoform nTRIP6. TRIP6 is translated from the first AUG by regular translation initiation. nTRIP6 is translated from a downstream in-frame AUG. The protein isoforms have different functions. While TRIP6 is a cytoplasmic regulator of the cytoskeleton, nTRIP6 is located in the nucleus and acts as a co-factor in transcription. The different sub-cellular localisation is caused by an nuclear export signal (NES) that reside between TRIP6-AUG and nTRIP6-AUG (e.g. mouse)/CUG (primates). The authors identify a small open reading frame just upstream AUG2 (internal upstream ORF, iuORF). Analysis of AUG-mutation experiments and translation-initiation interference with PNAs leads to the conclusion that translation from the AUG2 resulting in nTRIP6 is the result of leaky scanning and possibly in conditions of activated mTORC1 signalling through translation re-initiation of post iuORF ribosomes. Thereby mTORC1 activation would stimulate the expression of nTRIP6. Finally, an attempt is made to extend the findings and regulatory principles to other genes harbouring similar iuORFs.

The finding of conserved sORF/iuORFs is interesting and research to reveal their function may generate new knowledge about translation-regulatory mechanisms and additional functions of protein isoforms.

Major comments

1) The text needs improvement of style, grammar and logic flow of findings and arguments.

2) Abstract

Needs clearer writing:

We have identified etc... Why mystify the cis-regulatory element? Please consider mentioning the sORFs here. It now sounds like the authors discovered sORFs but this research is going on since the 1980 (e.g. by Marilyn Kozak).

Please reconsider the use of "dynamics" instead of just writing what it does, delaying differentiation.

3) Introduction

To state that regulation of proteoforms is poorly understood is an overstatement and does not reflect the literature. "partially understood" would be better.

Results

3) nTRIP6 is generated by alternative translation at an internal AUG & Figure 1:

We therefore predicted that etc... The lab already showed this and therefore it would suffice to just state that nTRIP6 goes nuclear because it lacks the N-terminal part with the NES.

This whole part needs rewriting and can be condensed considerably by avoiding duplications. E.g. at page 5 line 2-3 the NES is mentioned, and that comes back in line 19-20, and then it is mentioned again at page 6 line 6-7. Three times the same message!

4) Maybe good to mention the optimal Kozak consensus (G/A NN) G/A NN AUG G.

5) Fig 1B/C/D inconsistent labelling of control blots. B and C are cotransfected eGFP vectors, D is GR (glucocorticoid receptor?). Please be precise and transparent in figure labelling.

6) Data from Fig 1B and Fig 3B show what is to be expected, if you remove/block upstream AUG1/iuORF-AUG you gain translation from AUG2 just by making more ribosomes available for translation initiation at that site. Not surprising - considering the relative strength of two successive AUG1 and iuORF-AUG the expression of nTRIP6 is very low.

Hence a major point of worry is that the endogenous expression of nTRIP6 seems to be very low. It is a protein of reasonable size, and the antibody seems to work. Therefore, the observed difference in TRIP6 and mTRIP6 is probably real. The authors argue that transcription factors are generally lower expressed than proteins with more structural roles. However, even taking this into account we are dealing with very low expression and the question is whether the subtle regulation observed is physiological meaningful.

Now the authors will argue against my scepticism by referring to the invitro differentiation experiments using C2C12 cells. But really addressing this question would need an in vivo model system. Which, I acknowledge, is beyond of the scope of this manuscript (but the limitations of the invitro experiments can be discussed).

7) The layout of the figures is crowded. Also, it would really improve the reading if the authors would properly label the panel. Mention the cell types used, properly naming the loading/transfection controls.

The mTOR pathway transiently stimulates nTRIP6 translation during myogenesis - Figure 2

8) Concerning the screening of Medaka kinases. Maybe trivial, but why Medaka?

9) Why was nTRIP6/TRIP6 used and not nTRIP6/loading control (TRIP6 and nTRIP6 functions are independent)?

10) Figure 2A-B: IGF-1 was used to stimulate Akt-mTORC1 pathway. However, there seem not to be an increase in p-S6K. Therefore, the conclusion cannot be drawn that mTORC1 stimulates nTRIP6 expression. Often P-S6/S6 are easier to analyse. The authors should provide evidence that mTORC1 is affected by IGF-1 in this experiment. Also, here the expression of nTRIP6 is incredible low.

In addition, the experiment should be pared with mTORC1 inhibition (rapalog e.g. everolimus) to show the opposite regulation.

11) Fig 2C: Very, very low nTRIP6-HA expression. I do not see how to quantify this. Mistake in labelling of D "n" is missing in TRIP6/TRIP6. Quantification should be done compared to loading control. The TRIP6 signal is too strong, beyond linear quantification. Also, a downstream mTORC1 signalling target should be added to monitor the effect of RHEB, e.g. P-S6K/S6K or P-S6/S6 or P-4EBP1/4EBP1. Often using P-S6/S6 or P-4EBP1/4EBP1 works better.

In my opinion this experiment should be repeated or redesigned. It may be beyond technical possibility because of the extreme low expression of nTRIP6, but showing that nTRIP6 expression is be reduced upon mTORC1 inhibition would be informative.

12) For the whole figure it would really improve the reading if the authors would properly label the panels. Mentioning the cell types used, properly naming the loading/transfection controls. Why use alpha and beta for p values in panels D and F and not just label with the P values like in B and H?.

A short internal upstream open reading frame (iuORF) represses nTRIP6 translation - Figure 3

13) The authors should elaborate more on EV4 here, because it helps to explain and put into context the experimental data: EV4 shows or indicates that:

The mRNA structure is evolutionary conserved.

AUG1 is in a sub-optimal Kozak context (A/G NN AUG t/c) and prone to some leaky scanning.

iuORF-AUGs are in optimal or sub-optimal context (G/A/c NN AUG G/a) and likely to be translated by the ribosomes leaking over AUG1.

What are called AUG2s are not always AUGs, they are either AUG2 in sub-optimal Kozak consensus (A/G NN AUG t/c) or CUGs in optimal context (A NN CUG G), interestingly the latter are all primates. (except Zebrafish, which deviates by having iuORF and AUG2 in non-Kozak).

The sORF is rather long 78 to 192 nt = 26 to 64 codons.

The intercistronic gap between iuORF and AUG/CUG2 is short, 5 or 14 or 17 nucleotides.

Altogether the length of the iuORF and the small intercistronic spacer predicts a very inefficient re-initiation. The longer the sORF the less likely post-termination will produce scanning competent ribosomes, and these post-iuORF ribosomes must be reloaded with the ternary complex providing the initiator tRNA-Met, and this takes time. This has been investigated in depth by Marylin Kozak and others. This explains the very low endogenous expression of nTRIP6.

This in line with the experiments and the authors conclude that the iuORF represses translation from AUG2.

Instead of aiming to understand the details of nTRIP6 translational regulation and the mechanism of mTORC1 mediated upregulation the authors divert to other mRNAs with similar features (page 9 Figure 1D-L). See my suggestion below.

14) Figure 1 I and K:

The short/long isoform quantification is not appropriate: the TRIP6 signal is too strong and beyond the possibility of linear quantification. Short isoform/loading control should be used instead. Again, here the worry about the extreme low expression of the smaller isoforms.

In addition, Western blots should be included showing change in mTORC1 signalling, e.g. (P-)S6K, S6 or 4EBP1.

15) Page 10: The conclusion that "activation of mTORC1 ... increased the expression of ... short proteoform...without affecting the long proteoform" is premature. In several panels the actual prove that mTORC1 signal is altered is missing ((P-)S6K, S6 or 4EBP1 blots).

16) Please improve the figure layout with properly labelling of the panel. Mention the cell types used, properly naming the loading/transfection controls.

17) Page 10:Concerning re-initiation as the mechanism of mTORC1 mediated upregulation of short isoforms.

The authors could have performed experiments to address this question. Important because considering the architecture of the mRNA with relative long iuORF and small intercistronic spacer re-initiation is predictably inefficient.

My interpretation is that the iuORF acts like a filter, filtering out ribosomes that scan leaky over AUG1, and preventing translation initiation at AUG2. Meaning that translation from AUG2 into e.g. nTRIP6 can only occur by leaky scanning over AUG1 and iuORF and that is why the levels of nTRIP6 are very low. Activation of mTORC1 will result in more ribosome loading of the TRIP6-mRNA and result is just a bit more leaky scanning and translation at AUG2.

Relatively simple experiments can shed light on the involved mechanism, using transient expression of tagged TRIP6 constructs.

- optimise iuORF-AUG by placing it into A/G NN A/G NN AUG G context: will force translation at this site, diminishing leaky scanning and only potentially allowing for re-initiation.
- eventually weakening iuORF-AUG.
- shortening of the intercistronic spacer (e.g. 4 nt) and lengthening of the spacer, e.g. 30 nt, 60 nt, 120 nt: this will reveal whether re-initiation is really relevant for wt nTRIP6 regulation.

Referee #2:

Overall, the authors have identified a new type of regulatory element found within coding regions, internal upstream ORFs (iuORFs) that antagonize protein production from internal ORFs (iORFs). The authors demonstrated regulation of iORFs by iuORF sequences for TRIP6, Kctd20 and Tfdp1. The authors present data that altering mTORC signaling alters the levels of iORF translation when iuORFs are upstream of the iORF translation start site. Furthermore, treatment of C2C12 cells with AUG2 PNA leads to decreased nTRIP6 protein and to mRNA upregulation of early differentiation gene Myog and decreased expression of late differentiation genes like Tnni2 and MYH3, providing evidence that the internal AUG is critical for nTRIP6 protein expression and myoblast differentiation.

One major concern is that the authors have not shown that there is only one RNA isoform expressed from endogenous or transfected TRIP6 (and for transfected Kctd20 and Tfdp1). It is possible there is a promoter element downstream of the AUG1 giving rise to transcripts containing only the iuORF and iORF but in fact they are the uORF and main ORF of this alternative second transcript. See (Akirtava .. McManus 2022) in response to (Xue .. Barna 2015) for an example where knowledge of exactly where RNA transcripts begin influences interpretation of experiments suggesting translation regulation. Northern blots, 5' RACE and RNA transfections would better support that there are not alternative RNA isoforms. For candidates mentioned in Fig3C and 3D, CAGE data could be analyzed to determine whether there are transcription start sites downstream of cTIS but upstream of the iuTIS.

Another concern is the specificity of the AUG2 PNA used in Figure 1. If the phenotype presented in Figure 1H can be rescued by transfection of nTRIP6 that is not targeted by AUG2 PNA, this would provide strong evidence AUG2 PNA is not causing off target effects. Genome editing to mutate AUG2 in TRIP6 would also confirm that endogenous nTRIP6 translated from AUG2 is critical for myoblast differentiation.

Authors claim that because nTRIP6 protein levels increase after mutating the main ORF AUG1, that the nTRIP6 observed in wild-type constructs is due to leaky scanning. However, it's equally possible the nTRIP6 from wild-type constructs is from an internal iRES and nTRIP6 protein levels increase through leaky scanning following AUG1 mutation. If the authors engineered a reporter that prevented scanning (using structured RNA elements as in Jang & Paek 2016) and showed decreased nTRIP6 protein levels in reporters that prevented scanning that would support the hypothesis that nTRIP6 is translated through leaky scanning.

The authors often overstate their conclusions concerning TRIP6 protein levels, for example following mTORC1 inhibitor Everolimus treatment of C2C12 cells, the authors state "increase in mTORC1 activity promotes the increase in nTRIP6 translation". However, the authors do not show direct evidence of changes in TRIP6 translation, for example by using ribosome profiling, and present western blots indicating decrease of nTRIP6 protein levels. Post-translational regulation can't be ruled out, including post translational modifications affecting protein stability following mTOR activation, etc. Similarly, authors claim that because AUG2 PNA reduces nTRIP6 in C2C12 differentiation assays that the increase of nTRIP6 during differentiation is due to increases in translation initiation. However, it is still possible the increase of nTRIP6 during differentiation is due to post translational modifications affecting protein stability during myoblast differentiation. There is no direct evidence that there is higher translation of nTRIP6 during myoblast differentiation, would need ribosome profiling data. The AUG2 PNA experiment indicates that nTRIP6 translation is occurring from the internal AUG.

Authors claim that AUG2 PNA experiments alter the short but not long proteoform of TRIP6. However if the long proteoform is 50x more highly expressed than the short proteoform, a doubling of the short proteoform should only increase the long isoform by 2% and won't be detectable through western blotting.

The authors state the main TFDP1 proteoform was not expressed by the mAUG1 reporter (Fig 3H), but there is a band in that

lane indicating there could be translation from the ACG as a translation start site. ACG is a noncanonical translation start site reported by others (Ingolia et al 2011).

There is a new proteoform (band just below the strongest TRIP6 band) seen in AUG2 PNA treated cells undergoing differentiation (Fig 1D) but not commented on. Is there a predicted translation start that gives rise to this proteoform?

While analysis of ribosome profiling experiments was used, no data from that analysis is shown. Metaplots of ribosome profiling data for the candidates described in Figure3C and Figure3D would convey the results more strongly, expecting to see out of frame ribosome profiling reads in the iuORF region.

The evolutionary analysis in EV4 is intriguing and could be expanded to the other candidate genes in Fig 3C and 3D. Is this selection stronger than in coding regions of other similarly conserved genes?

Perhaps because short proteoforms are expressed at much lower level than the long proteoforms, some of the interpretations of the western blot data don't line up with the data presented. For example in Figure 1C, the nTRIP6 band looks to be at similar levels in the Misp and AUG2 lanes for the wt reporter, is there quantification for this data? Also, in Figure 3I, there seems to be more long proteoform of Kctd20 in the rightmost lane compared to the left, for accurate quantification it will be important that the top bands are not overexposed, is it possible to show the other 2 replicates in the supplement?

Figure 2D y-axis should read "nTRIP6/TRIP6"

Referee #3:

General:

Fettig et al, propose that expression of a truncated form of TRIP6 (nTRIP6) arises via leaky scanning by the ribosome, resulting in translation initiation at a downstream AUG. Furthermore, it is proposed that there is a short open reading frame within the coding sequence which has a role in suppressing synthesis of the truncated nTRIP6. Finally, the authors propose that these internal short open reading frames have a more general role in regulating synthesis of N-terminally truncated proteoforms. This intricate mode of regulation, which mirrors the role that upstream open reading frames can have in regulating translation of the main open reading, would be interesting and to my knowledge novel. However, there are a number of concerns which in my opinion questions these conclusions.

Major:

1. On page 5 it is argued that the results from Fig 1A,B (where mutation of the AUG of the main ORF results in increased expression of nTRIP6) strongly suggests that nTRIP6 arises from alternative translation initiation after leaky scanning. This is an over-interpretation of the data as it only shows that protein synthesis can arise from the downstream AUG. Expression of the downstream AUG could also be a consequence of synthesis of mRNA from an alternative transcription start site. I had a quick look in the Fantom5 data base for TRIP6 and it looks like there are several downstream transcription start sites indicated within the open reading frame. Accordingly, it is necessary to show that expression of alternative mRNA isoforms does not explain the results obtained in the C2C12 model. Moreover, using ectopic expression does not exclude that minor transcript isoforms could arise due to cryptic promoters. Therefore, it is essential to also ensure that there are no mRNA isoforms that lack the first AUG. There are several ways to accomplish this, one being in vitro synthesis of the mRNA, isolation of the full length transcript followed by transfection into cells. Furthermore, CAGE analysis (which may already exist in the Fantom5 data base) would also reveal the existence of alternative transcription start sites. Several approaches will be required to rule out the potential contribution of alternative transcript isoforms.
2. The data in figure 1D-E does not show a convincing increase in nTRIP6 as normalization appears to have been done relative to a band approaching saturation. Also, expression of nTRIP6 appears to correlate with signal of the loading control. The results obtained in the study of Norizadeh Abbariki et al. 2021 shows a much more convincing induction of nTRIP6. Also, there is no data on expression of nTRIP6 without any PNA treatment, which is important to determine whether PNA has a non-specific effect on nTRIP6 expression.
3. The rationale of the PNA experiment needs to be better explained (the assumption seems to be that the PNA would only affect translation initiation?). Furthermore, as PNA has the potential to inhibit both transcription and translation it is necessary to ensure that it does not affect expression of mRNA isoforms (similar to point 1 above). Also, it is typically required to perform a rescue by e.g. altering the PNA binding site in the target mRNA to show that this leads to a loss of activity.
4. In figure 2A the induction of phosphorylation of p70S6K is modest. Activation of the mTOR pathway is typically much stronger. There is no assessment of activity of the mTOR pathway in figure 2C (and similar experiments in figure 3) which uses an alternative approach to activate mTOR. As mTOR has been reported as a regulator of transcription, assessing mRNA isoform expression as described under point 1 is essential.
5. It appears that the data in figure 1D and 2G for the control conditions should be very similar. In the shown figures they look very different. Moreover, the loading in figure 2G seems very uneven.
6. EV3 shows a western blotting for ATF4 with many bands. Without a positive control showing which of these bands corresponds to ATF4 (e.g. by treating cells with thapsigargin) the effects on ATF4 cannot be deduced.

7. Figure 3 shows a number of experiments that may be explained by alternative transcription start sites giving rise to alternative transcript isoforms, rather than leaky scanning. It is therefore essential to determine the repertoire of transcription start sites in these experiments (as discussed under point 1).

8. Quantifications of western blots in figure 3 where the long proteomforms are quantified at saturation and compared to lowly expressed truncated protein cannot reveal their relative expression.

Minor

1. There is no indication of which phosphorylation sites is assessed in western blotting figures.

Referee #1:

Major comments

1) *The text needs improvement of style, grammar and logic flow of findings and arguments.*

The manuscript has now been rewritten as a full article, and no more as a short report. During the rewriting process, we have generally improved these aspects of the text.

2) Abstract

Needs clearer writing:

We have identified etc... Why mystify the cis-regulatory element? Please consider mentioning the sORFs here. It now sounds like the authors discovered sORFs but this research is going on since the 1980 (e.g. by Marilyn Kozak).

Please reconsider the use of "dynamics" instead of just writing what it does, delaying differentiation.

We have modified the abstract to incorporate the new results that we present. As recommended by the reviewer, we now also immediately mention sORFs. However, specifying "what nTRIP6 does" would require too much space, as it is not simply delaying differentiation. Therefore, due to abstract length limitation and considering that the function of nTRIP6 during myogenesis is not the main focus of the manuscript, we have not modified this sentence.

3) Introduction

To state that regulation of proteoforms is poorly understood is an overstatement and does not reflect the literature. "partially understood" would be better.

We agree with the reviewer and have corrected this statement.

Results

3) *nTRIP6 is generated by alternative translation at an internal AUG & Figure 1:*

We therefore predicted that etc... The lab already showed this and therefore it would suffice to just state that nTRIP6 goes nuclear because it lacks the N-terminal part with the NES.

This whole part needs rewriting and can be condensed considerably by avoiding duplications. E.g. at page 5 line 2-3 the NES is mentioned, and that comes back in line 19-20, and then it is mentioned again at page 6 line 6-7. Three times the same message!

As recommended, we have now entirely rewritten this part, since we now also present mRNA analysis results.

4) *Maybe good to mention the optimal Kozak consensus (G/A NN) G/A NN AUG G.*

Indeed, it makes a lot of sense to mention the optimal Kozak sequence, in particular in view of the new results we present in Figures 2 and 5. We thank the reviewer for this suggestion.

5) Fig 1B/C/D inconsistent labelling of control blots. B and C are cotransfected eGFP vectors, D is GR (glucocorticoid receptor?). Please be precise and transparent in figure labelling.

We have corrected all the western blots labelling to be more precise.

6) Data from Fig 1B and Fig 3B show what is to be expected, if you remove/block upstream AUG1/iuORF-AUG you gain translation from AUG2 just by making more ribosomes available for translation initiation at that site. Not surprising - considering the relative strength of two successive AUG1 and iuORF-AUG the expression of nTRIP6 is very low.

Hence a major point of worry is that the endogenous expression of nTRIP6 seems to be very low. It is a protein of reasonable size, and the antibody seems to work. Therefore, the observed difference in TRIP6 and mTRIP6 is probably real. The authors argue that transcription factors are generally lower expressed than proteins with more structural roles. However, even taking this into account we are dealing with very low expression and the question is whether the subtle regulation observed is physiological meaningful.

Now the authors will argue against my scepticism by referring to the invitro differentiation experiments using C2C12 cells. But really addressing this question would need an *in vivo* model system. Which, I acknowledge, is beyond of the scope of this manuscript (but the limitations of the invitro experiments can be discussed).

We do agree that the expression of nTRIP6 is very low relative to that of TRIP6. However, this is only a relative quantification of nTRIP6, and thus we do not see how it is possible to conclude as to the physiological relevance of nTRIP6 based on this relative quantification. We have already documented the function of endogenous nTRIP6 using various loss of function approaches (e.g. Dieffenbacher et al 2014). In particular we have shown that endogenous nTRIP6 in myoblasts acts as a co-repressor for MEF2C (Kemler et al 2016). Furthermore, we have addressed the relevance of nTRIP6 in an *in vivo* model of regeneration in the mouse (Norizadeh Abbariki et al 2021). We acknowledge the fact that focus on *in vitro* experiments in our manuscript. However, as mentioned by the reviewer, addressing the *in vivo* relevance of the mechanism that we describe is far beyond of the scope of this manuscript.

7) The layout of the figures is crowded. Also, it would really improve the reading if the authors would properly label the panel. Mention the cell types used, properly naming the loading/transfection controls.

We agree that the layout of our figures may have been confusing. Therefore, all the figures have been modified accordingly.

The mTOR pathway transiently stimulates nTRIP6 translation during myogenesis - Figure 2

8) Concerning the screening of Medaka kinases. Maybe trivial, but why Medaka?

When one overexpresses a mammalian kinase in mammalian cells, it is not necessarily active if the upstream activating signal is not provided. However, medaka

kinases overexpressed in mammalian cells are active. The reason for this is not known but this property has already been used (Chen, et al. (2014) EMBO Rep 15, 1254-1267; Zhang et al (2015) BMC Biotechnol 15:92) and might have to do with putting the kinase in a much higher temperature environment. Luckily, the overexpression library was available so we used it.

9) Why was nTRIP6/TRIP6 used and not nTRIP6/loading control (TRIP6 and nTRIP6 functions are independent)?

We originally presented the nTRIP6/Trip6 ratio for transfection experiments, considering that normalizing to an endogenous protein as a loading control may not account for potential variations in transfection efficiency. However, now, in all figures we present TRIP6 and nTRIP6 expression relative to either the loading control or to a transfection control.

10) Figure 2A-B: IGF-1 was used to stimulate Akt-mTORC1 pathway. However, there seem not to be an increase in p-S6K. Therefore, the conclusion cannot be drawn that mTORC1 stimulates nTRIP6 expression. Often P-S6/S6 are easier to analyse. The authors should provide evidence that mTORC1 is affected by IGF-1 in this experiment. Also, here the expression of nTRIP6 is incredible low.

In addition, the experiment should be paired with mTORC1 inhibition (rapalog e.g. everolimus) to show the opposite regulation.

We agree with the reviewer that IGF-1 treatment did not promote a very strong activation of mTORC1. This is probably due to the relatively high starting / basal mTORC1 activity in proliferating C2C12 myoblasts, as can be seen in Figure 4F. However, this activation is significant, as shown by the quantification that we now present in the new Figure 4C. We have now added results showing that the mTORC1 inhibitor Everolimus inhibits the IGF-1-induced increase in nTRIP6 levels, confirming the involvement of mTORC1 in the effect of IGF-1.

11) Fig 2C: Very, very low nTRIP6-HA expression. I do not see how to quantify this. Mistake in labelling of D "n" is missing in TRIP6/TRIP6. Quantification should be done compared to loading control. The TRIP6 signal is too strong, beyond linear quantification. Also, a downstream mTORC1 signalling target should be added to monitor the effect of RHEB, e.g. P-S6K/S6K or P-S6/S6 or P-4EBP1/4EBP1. Often using P-S6/S6 or P-4EBP1/4EBP1 works better.

In my opinion this experiment should be repeated or redesigned. It may be beyond technical possibility because of the extreme low expression of nTRIP6, but showing that nTRIP6 expression is reduced upon mTORC1 inhibition would be informative.

We now provide new results in Figure 4D,E. As for the other figures, the nTRIP6 and TRIP6 expression levels are now normalized to the loading / transfection control, and the quantification has been made within the linear range of detection. Unfortunately, we were not able to detect an increase in the phosphorylation of endogenous p70S6K upon transfection of the constitutively active RHEB construct. This is

undoubtedly due to the transfection efficiency which is notoriously very low in C2C12 myoblasts. This is the reason why we have studied the effect of RHEB on a co-transfected, tagged *Trip6* CDS construct. However, we do now show that the increase in nTRIP6 expression upon RHEB transfection is inhibited by Everolimus (new Figure 4D,E), confirming the involvement of mTORC1.

12) *For the whole figure it would really improve the reading if the authors would properly label the panels. Mentioning the cell types used, properly naming the loading/transfection controls. Why use alpha and beta for p values in panels D and F and not just label with the P values like in B and H?*

The whole figure (new Figure 4) has been redesigned and all of these points raised have now been addressed.

A short internal upstream open reading frame (iuORF) represses nTRIP6 translation - Figure 3

13) *The authors should elaborate more on EV4 here, because it helps to explain and put into context the experimental data:*

EV4 shows or indicates that:

The mRNA structure is evolutionary conserved.

AUG1 is in a sub-optimal Kozak context (A/G NN AUG t/c) and prone to some leaky scanning.

iuORF-AUGs are in optimal or sub-optimal context (G/A/c NN AUG G/a) and likely to be translated by the ribosomes leaking over AUG1.

What are called AUG2s are not always AUGs, they are either AUG2 in sub-optimal Kozak consensus (A/G NN AUG t/c) or CUGs in optimal context (A NN CUG G), interestingly the latter are all primates. (except Zebrafish, which deviates by having iuORF and AUG2 in non-Kozak).

The sORF is rather long 78 to 192 nt = 26 to 64 codons.

The intercistronic gap between iuORF and AUG/CUG2 is short, 5 or 14 or 17 nucleotides.

Altogether the length of the iuORF and the small intercistronic spacer predicts a very inefficient re-initiation. The longer the sORF the less likely post-termination will produce scanning competent ribosomes, and these post-iuORF ribosomes must be reloaded with the ternary complex providing the initiator tRNA-Met, and this takes time. This has been investigated in depth by Marilyn Kozak and others. This explains the very low endogenous expression of nTRIP6.

This in line with the experiments and the authors conclude that the iuORF represses translation from AUG2.

Instead of aiming to understand the details of nTRIP6 translational regulation and the mechanism of mTORC1 mediated upregulation the authors divert to other mRNAs with similar features (page 9 Figure 1D-L). See my suggestion below.

We thank the reviewer for these suggestions. Now that we have rewritten the manuscript as a full article and no more as a short report, we were able to address more of these issues. We indeed now comment more appropriately Figure EV4 (now Figures EV2 and EV5). Furthermore, following the reviewer's suggestion, we have performed new experiments (see below).

14) Figure 1 I and K:

The short/long isoform quantification is not appropriate: the TRIP6 signal is too strong and beyond the possibility of linear quantification. Short isoform/loading control should be used instead. Again, here the worry about the extreme low expression of the smaller isoforms.

In addition, Western blots should be included showing change in mTORC1 signalling, e.g. (P-)S6K, S6 or 4EBP1.

15) Page 10: *The conclusion that "activation of mTORC1 ... increased the expression of ... short proteoform...without affecting the long proteoform" is premature. In several panels the actual prove that mTORC1 signal is altered is missing ((P-)S6K, S6 or 4EBP1 blots).*

We have now quantified the long and short proteoforms, within the linear range of detection relatively to the loading control (new Figure 8). As discussed above in point 11, the transfection efficiency in C2C12 is too low for the effect of the transfected RHEB construct to be detectable on endogenous mTORC1 substrates. However, given that the effect of the constitutively active RHEB construct on nTRIP6 expression is inhibited by Everolimus (Figure 4D), it is very likely that the increase in the expression of the short KCTD20 proteoform upon co-transfection of the RHEB construct is also mediated by mTORC1. However, given that we do not provide formal evidence for this, we have now modulated our concluding statements.

16) *Please improve the figure layout with properly labelling of the panel. Mention the cell types used, properly naming the loading/transfection controls*

The Figure has been modified accordingly to provide all this information.

17) Page 10: *Concerning re-initiation as the mechanism of mTORC1 mediated upregulation of short isoforms.*

The authors could have performed experiments to address this question. Important because considering the architecture of the mRNA with relative long iuORF and small intercistronic spacer re-initiation is predictably inefficient.

My interpretation is that the iuORF acts like a filter, filtering out ribosomes that scan leaky over AUG1, and preventing translation initiation at AUG2. Meaning that translation from AUG2 into e.g. nTRIP6 can only occur by leaky scanning over AUG1 and iuORF and that is why the levels of nTRIP6 are very low. Activation of mTORC1 will result in more ribosome loading of the TRIP6-mRNA and result is just a bit more leaky scanning and translation at AUG2.

Relatively simple experiments can shed light on the involved mechanism, using transient expression of tagged TRIP6 constructs.

- optimise iuORF-AUG by placing it into A/G NN A/G NN AUG G context: will force translation at this site, diminishing leaky scanning and only potentially allowing for re-initiation.

- eventually weakening iuORF-AUG.

- shortening of the intercistronic spacer (e.g. 4 nt) and lengthening of the spacer, e.g. 30 nt, 60 nt, 120 nt: this will reveal whether re-initiation is really relevant for wt nTRIP6 regulation.

We thank the reviewer for these suggestions. We now present a set of experiments which address the mechanism of this regulation. In particular, we present novel evidence confirming leaky scanning at AUG1 (new Figure 2). Using a translation initiation assay based on a ribosome profiling protocol, we confirm an increased translation initiation at AUG2 during myogenesis, while translation initiation at AUG1 and at the iuORF AUG do not significantly vary (new Figure 3C). New results show that the expression of nTRIP6 most likely occurs after leaky scanning at both AUG1 and at the iuORF AUG, and that the expression of nTRIP6 increases when the distance between the stop codon of the iuORF and AUG2 is increased, confirming that the iuORF prevents re-initiation at AUG2 (new Figure 5). Furthermore, we show that both reducing and increasing the length of the intercistronic spacer abolishes the increased expression of nTRIP6 upon RHEB transfection (new Figure 6A,B), confirming the relevance of re-initiation. Finally, we show using the translation initiation assay that this increased initiation at AUG2 is inhibited by Everolimus, while initiation at AUG1 and at the iuORF AUG are not affected (new Figure 6C). Indeed, these new results, as the reviewer suggested, do lead to the interpretation that translation initiation at AUG2 nTRIP6 can only occur by leaky scanning over AUG1 and the iuORF, resulting in very low levels of nTRIP6. However, our new results rather fit with the interpretation that activation of mTORC1 favours re-initiation at AUG2 after the iuORF.

Referee #2:

Overall, the authors have identified a new type of regulatory element found within coding regions, internal upstream ORFs (iuORFs) that antagonize protein production from internal ORFs (iORFs). The authors demonstrated regulation of iORFs by iuORF sequences for TRIP6, Kctd20 and Tfdp1. The authors present data that altering mTORC signaling alters the levels of iORF translation when iuORFs are upstream of the iORF translation start site. Furthermore, treatment of C2C12 cells with AUG2 PNA leads to decreased nTRIP6 protein and to mRNA upregulation of early differentiation gene Myog and decreased expression of late differentiation genes like Tnni2 and MYH3, providing evidence that the internal AUG is critical for nTRIP6 protein expression and myoblast differentiation.

One major concern is that the authors have not shown that there is only one RNA isoform expressed from endogenous or transfected TRIP6 (and for transfected Kctd20 and Tfdp1). It is possible there is a promoter element downstream of the AUG1 giving rise to transcripts containing only the iuORF and iORF but in fact they are the uORF and main ORF of this alternative second transcript. See (Akirtava .. McManus 2022) in response to (Xue .. Barna 2015) for an example where knowledge of exactly where RNA transcripts begin influences interpretation of experiments suggesting translation regulation. Northern blots, 5' RACE and RNA transfections would better support that there are not alternative RNA isoforms. For candidates mentioned in Fig3C and 3D, CAGE data could be analyzed to determine

whether there are transcription start sites downstream of cTIS but upstream of the iuTIS.

We now provide an mRNA analysis (new Figure EV1 and Appendix Figure S1) that confirms that although *Trip6* mRNA variants do indeed exist, they cannot account for the expression of nTRIP6. Furthermore, Figure 1D (previously Figure 1C) shows the proteoforms translated from *Trip6* cDNA in an in vitro assay. Thus, it seems highly unlikely that nTRIP6 is translated from an mRNA transcribed from an internal promoter that would also be recognized by T7 polymerase which was used in this assay. For the other candidates KCTD20 and TFDP1, we cannot formally exclude the existence of alternative mRNAs that would arise from putative transcription start sites downstream of the cTIS and upstream of the iuTIS in the transfected CDS. Consequently, we cannot formally exclude that such mRNAs would be translated into the short KCTD20 and TFDP1 proteoforms. For this reason, we have now modified our conclusions.

Another concern is the specificity of the AUG2 PNA used in Figure 1. If the phenotype presented in Figure 1H can be rescued by transfection of nTRIP6 that is not targeted by AUG2 PNA, this would provide strong evidence AUG2 PNA is not causing off target effects. Genome editing to mutate AUG2 in TRIP6 would also confirm that endogenous nTRIP6 translated from AUG2 is critical for myoblast differentiation.

We did not perform any phenotype rescue experiment by overexpression of nTRIP6 as it would have led to overexpression at much higher levels than the endogenous protein, in particular during the proliferation phase, which would have most likely affected differentiation. Thus, any effect on the phenotype would have been very problematic to interpret regarding the PNA specificity.

We now present new results to validate the PNA (Figure 1F; Figure EV3).

We indeed have considered genome editing to mutate AUG2. However, we have observed that mutating AUG2 to a non-initiating codon also increases the expression of lower molecular weight isoforms (visible on the longer exposed western blot that we present in Figure 1B). We do not know what the effect of these shorter isoforms might be. Thus, the effect of mutating AUG2 by genome editing on myoblast differentiation would have been very difficult to interpret.

Importantly, the PNA targeting nTRIP6 AUG perfectly phenocopies the effect of blocking the function of endogenous nTRIP6 on differentiation and fusion (Norizadeh Abbariki et al 2021). We now discuss this strong evidence for the specificity of the effect of the PNA, since we have now rewritten the manuscript as a full article, which includes a discussion section.

Authors claim that because nTRIP6 protein levels increase after mutating the main ORF AUG1, that the nTRIP6 observed in wild-type constructs is due to leaky scanning. However, it's equally possible the nTRIP6 from wild-type constructs is from an internal iRES and nTRIP6 protein levels increase through leaky scanning following AUG1 mutation. If the authors engineered a reporter that prevented scanning (using structured RNA elements as in Jang & Paek 2016) and showed decreased nTRIP6 protein levels in reporters that prevented scanning that would support the hypothesis that nTRIP6 is translated through leaky scanning.

We thank the reviewer for this suggestion. We now present new results in favour of the leaky scanning hypothesis. Indeed, the introduction of a stable hairpin just after the cap strongly inhibits the expression of both TRIP6 and nTRIP6 to the same extent (Figure 2A-D). Moreover, strengthening the Kozak context of AUG1 slightly decreases nTRIP6 expression, however not statistically significantly. Conversely, weakening the AUG1 Kozak context decreases the expression of TRIP6 and strongly increases that of nTRIP6 (Fig 2E-G).

The authors often overstate their conclusions concerning TRIP6 protein levels, for example following mTORC1 inhibitor Everolimus treatment of C2C12 cells, the authors state "increase in mTORC1 activity promotes the increase in nTRIP6 translation". However, the authors do not show direct evidence of changes in TRIP6 translation, for example by using ribosome profiling, and present western blots indicating decrease of nTRIP6 protein levels. Post-translational regulation can't be ruled out, including post translational modifications affecting protein stability following mTOR activation, etc. Similarly, authors claim that because AUG2 PNA reduces nTRIP6 in C2C12 differentiation assays that the increase of nTRIP6 during differentiation is due to increases in translation initiation. However, it is still possible the increase of nTRIP6 during differentiation is due to post translational modifications affecting protein stability during myoblast differentiation. There is no direct evidence that there is higher translation of nTRIP6 during myoblast differentiation, would need ribosome profiling data. The AUG2 PNA experiment indicates that nTRIP6 translation is occurring from the internal AUG.

We now present new results which confirm that nTRIP6 translation is increased in an mTORC1-dependent manner during myoblast differentiation. In a translation initiation assay based on a ribosome profiling protocol, translation initiation at AUG2 was increased during myogenesis, while translation initiation at AUG1 and at the iuORF AUG did not significantly vary (new Figure 3C and 6C). This increased initiation at AUG2 was inhibited by Everolimus, while initiation at AUG1 and at the iuORF AUG was not affected (new Figure 6C).

Authors claim that AUG2 PNA experiments alter the short but not long proteoform of TRIP6. However if the long proteoform is 50x more highly expressed than the short proteoform, a doubling of the short proteoform should only increase the long isoform by 2% and won't be detectable through western blotting.

We now present new results showing the effect of the PNA on TRIP6 and nTRIP6 expression (Figure 1F,G) in which we did not see any significant modification of TRIP6 expression. However, we do agree with the reviewer that a 2% change would not be detectable. For this reason, we have modified our claims in the text by now writing "...without significantly affecting the expression of TRIP6".

The authors state the main TFDP1 proteoform was not expressed by the mAUG1 reporter (Fig 3H), but there is a band in that lane indicating there could be translation from the ACG as a translation start site. ACG is a noncanonical translation start site reported by others (Ingolia et al 2011).

We thank the reviewer for raising this issue. To address it' we have now modified the text: "...corresponding to the usage of the annotated first AUG. Its expression was strongly reduced but not totally abolished when this AUG was mutated into a non-initiating codon, suggesting that in the context of this mRNA the ACG codon used worked as a weak initiation codon, as already reported (Ingiólia et al, 2011)."

There is a new proteoform (band just below the strongest TRIP6 band) seen in AUG2 PNA treated cells undergoing differentiation (Fig 1D) but not commented on. Is there a predicted translation start that gives rise to this proteoform?

We do now comment on this band. It most likely arises from a CUG initiation site in a relatively good Kozak context located 78 nucleotides downstream of AUG1.

The evolutionary analysis in EV4 is intriguing and could be expanded to the other candidate genes in Fig 3C and 3D. Is this selection stronger than in coding regions of other similarly conserved genes?

One of the filtration criteria for the identification of candidate genes was indeed the conservation of the most relevant features for short ORFs to repress initiation at the next downstream TIS (see Materials and Methods section and Table EV2). While we agree with the reviewer that a more comprehensive evolutionary analysis would be interesting, it goes beyond the scope of the current manuscript.

Perhaps because short proteoforms are expressed at much lower level than the long proteoforms, some of the interpretations of the western blot data don't line up with the data presented. For example in Figure 1C, the nTRIP6 band looks to be at similar levels in the Misp and AUG2 lanes for the wt reporter, is there quantification for this data? Also, in Figure 3I, there seems to be more long proteoform of Kctd20 in the rightmost lane compared to the left, for accurate quantification it will be important that the top bands are not overexposed, is it possible to show the other 2 replicates in the supplement?

We now present quantifications of all the Western blots. In the one presented in Figure 1C (now Figure 1D), there is a significant reduction of the nTRIP6 band intensity. For a clearer visualization, we have replaced this blot by one of the repetitions, in which the effect of the PNA was more visible.

In the original Figure 3, now Figure 8, the quantification which was based on the ratio between the short and the long isoforms is now relative to the loading control. There was a slight increase in the expression of the long isoform of KCTD20, as can be seen from the individual values depicted in the plot (Figure 8H). However, this increase was not as pronounced as that observed for the short isoform and was not significant. We have modified the text in the result section accordingly.

As indicated in the Materials and Methods section, all quantification was performed within the linear range of detection, i.e. before overexposure of the bands.

Figure 2D y-axis should read "nTRIP6/TRIP6"

We apologise for this mistake. We now present all western blot quantifications relative to the loading or to the transfection controls.

Referee #3:

General:

Fettig et al, propose that expression of a truncated form of TRIP6 (nTRIP6) arises via leaky scanning by the ribosome, resulting in translation initiation at a downstream AUG. Furthermore, it is proposed that there is a short open reading frame within the coding sequence which has a role in suppressing synthesis of the truncated nTRIP6. Finally, the authors propose that these internal short open reading frames have a more general role in regulating synthesis of N-terminally truncated proteoforms. This intricate mode of regulation, which mirrors the role that upstream open reading frames can have in regulating translation of the main open reading, would be interesting and to my knowledge novel. However, there are a number of concerns which in my opinion questions these conclusions.

Major:

1. *On page 5 it is argued that the results from Fig 1A,B (where mutation of the AUG of the main ORF results in increased expression of nTRIP6) strongly suggests that nTRIP6 arises from alternative translation initiation after leaky scanning. This is an over-interpretation of the data as it only shows that protein synthesis can arise from the downstream AUG. Expression of the downstream AUG could also be a consequence of synthesis of mRNA from an alternative transcription start site. I had a quick look in the Fantom5 data base for TRIP6 and it looks like there are several downstream transcription start sites indicated within the open reading frame. Accordingly, it is necessary to show that expression of alternative mRNA isoforms does not explain the results obtained in the C2C12 model. Moreover, using ectopic expression does not exclude that minor transcript isoforms could arise due to cryptic promoters. Therefore, it is essential to also ensure that there are no mRNA isoforms that lack the first AUG. There are several ways to accomplish this, one being in vitro synthesis of the mRNA, isolation of the full length transcript followed by transfection into cells. Furthermore, CAGE analysis (which may already exist in the Fantom5 data base) would also reveal the existence of alternative transcription start sites. Several approaches will be required to rule out the potential contribution of alternative transcript isoforms.*

We provide new results showing the existence of *Trip6* mRNA variants (new Figure EV1 and Appendix Figure S1). However, none of these variants can account for the expression of nTRIP6. Furthermore, Figure 1D (previously Figure 1C) shows that also in an *in vitro* assay a proteoform is translated from AUG2. It seems very unlikely that a cryptic promoter downstream of ATG1 would also be recognized by T7 polymerase which was used in this assay. In a new set of experiments, we show that the introduction of a stable hairpin just after the cap strongly inhibits the expression of both TRIP6 and nTRIP6 to the same extent, while, importantly, the mRNA level was not affected, as monitored by the expression of eGFP driven by an IRES in the same construct (Figure 2A-D). Together, our new results show that TRIP6 and nTRIP6 are translated from the same mRNA, nTRIP6 is not translated from an IRES, and leaky scanning at AUG1 enables translation of nTRIP6.

For the other candidates KCTD20 and TFDP1, we cannot formally exclude the existence of alternative transcripts. Therefore, we have modified our conclusions appropriately (see also our answer to point 7).

2. The data in figure 1D-E does not show a convincing increase in nTRIP6 as normalization appears to have been done relative to a band approaching saturation. Also, expression of nTRIP6 appears to correlate with signal of the loading control. The results obtained in the study of Norizadeh Abbariki et al. 2021 shows a much more convincing induction of nTRIP6. Also, there is no data on expression of nTRIP6 without any PNA treatment, which is important to determine whether PNA has a non-specific effect on nTRIP6 expression.

As mentioned in the Materials and Methods section, quantification was performed within the linear range of detection, i.e. before the exposure was saturated. We agree that the loading control is not perfectly even. However, the quantification of nTRIP6 expression shows an increase in the control conditions which is very similar to that reported in Norizadeh Abbariki et al. 2021.

We provide new results to validate the effect of the PNA (Figure 1F,G, Figure EV3). In particular, only the AUG2-targeting PNA, and not the mispaired control nor a random sequence PNA, reduced the expression of nTRIP6 as compared to untreated cells (Figure 1F,G), confirming that the PNA treatment per se does not affect the expression of nTRIP6.

3. The rationale of the PNA experiment needs to be better explained (the assumption seems to be that the PNA would only affect translation initiation?). Furthermore, as PNA has the potential to inhibit both transcription and translation it is necessary to ensure that it does not affect expression of mRNA isoforms (similar to point 1 above). Also, it is typically required to perform a rescue by e.g. altering the PNA binding site in the target mRNA to show that this leads to a loss of activity.

Indeed, PNAs can also bind to DNA and can therefore potentially inhibit transcription if targeted to regulatory regions of the gene. However, the AUG2 PNA that we used does not target such a region. Furthermore, we now provide new results showing that the AUG2 targeting PNA does not reduce, but surprisingly rather increases *Trip6* mRNA levels in C2C12 myoblasts (Figure EV3).

We did not perform a phenotype rescue experiment as ectopically expressing nTRIP6 would have resulted in higher levels than the endogenous protein, also during the proliferation phase, which would most likely have affected differentiation. Thus, any effect on the phenotype would have been very problematic to interpret in terms of specificity of the PNA. However, the PNA targeting nTRIP6 AUG perfectly phenocopies the effect of blocking the function of endogenous nTRIP6 on differentiation and fusion (Norizadeh Abbariki et al 2021). We now discuss this strong evidence for the specificity of the effect of the PNA, since we have now rewritten the manuscript as a full article, which includes a discussion section.

4. In figure 2A the induction of phosphorylation of p70S6K is modest. Activation of the mTOR pathway is typically much stronger. There is no assessment of activity of the mTOR pathway in figure 2C (and similar experiments in figure 3) which uses an alternative approach to activate mTOR. As mTOR has been reported as a regulator

of transcription, assessing mRNA isoform expression as described under point 1 is essential.

We agree with the reviewer that IGF-1 did not promote a very robust activation of mTORC1, as assessed by p70S6K phosphorylation. This probably stems from the relatively high starting / basal mTORC1 activity in proliferating C2C12 myoblasts, as can be seen in Figure 4F. However, this IGF-1-mediated stimulation of mTORC1, which is now quantified in Figure 4C, was sufficient to promote an increase in nTRIP6 expression. Indeed, we now show that the IGF-1-induced increase in nTRIP6 expression is inhibited by the mTORC1 inhibitor Everolimus (see new results in Figure 4A,B).

Transfection efficiency is notoriously very low in C2C12 myoblasts. For this reason, in the experiment presented in Figure 2C and 3 (now 2D and 8), we were unfortunately not able to detect an increase in the phosphorylation of endogenous p70S6K upon transfection of the constitutively active RHEB construct. However, we do now provide evidence that the RHEB construct indeed increases nTRIP6 expression via mTORC1, as the increase is inhibited by Everolimus (new Figure 4D,E).

As mentioned above, no *Trip6* mRNA variant can account for the expression of nTRIP6. Thus, the increase in nTRIP6 expression induced by mTORC1 cannot have occurred via a transcriptional effect, considering that TRIP6 levels were not significantly affected. Furthermore, we have reported that *Trip6* mRNA levels do not vary during C2C12 myoblast differentiation (Norizadeh Abbariki et al 2021), despite the variation in mTORC1 activity (Figure 4F,G).

5. It appears that the data in figure 1D and 2G for the control conditions should be very similar. In the shown figures they look very different. Moreover, the loading in figure 2G seems very uneven.

We do not fully understand what the reviewer is referring to here. On the contrary, the quantification shows that the data in Figure 1D (now 3A) and 2G (now 4H) for the control conditions are rather consistent. Of course, we have noticed variability in the absolute levels of TRIP6 and nTRIP6 from independent experiment to independent experiment, which is perhaps attributable to differences in the cell passage number of this very delicate cell line.

We do agree with the reviewer that the loading in Figure 2G (now 4H) is not even. However, in the quantification, the normalization of TRIP6 proteoforms expression to the loading control indicates that the trends are consistently the same.

6. EV3 shows a western blotting for ATF4 with many bands. Without a positive control showing which of these bands corresponds to ATF4 (e.g. by treating cells with thapsigargin) the effects on ATF4 cannot be deduced.

Since we could not acquire a better anti-ATF4 antibody and since these results are only accessory we have removed them from the manuscript.

7. Figure 3 shows a number of experiments that may be explained by alternative transcription start sites giving rise to alternative transcript isoforms, rather than leaky scanning. It is therefore essential to determine the repertoire of transcription start sites in these experiments (as discussed under point 1).

We cannot formally exclude the existence of alternative mRNAs that would be transcribed from the transfected plasmids encoding the *Kctd20* and *Tfdp1* CDSs at a putative cryptic promoter downstream of the first AUG but upstream of the short ORFs. Consequently, we cannot formally exclude that such mRNAs would be translated into the short KCTD20 and TFDP1 proteoforms. For this reason, we have now modified our conclusions.

8. Quantifications of western blots in figure 3 where the long proteoforms are quantified at saturation and compared to lowly expressed truncated protein cannot reveal their relative expression.

For all the western blots, we now present quantification of the long and short proteoforms relative to loading controls. Furthermore, as specified in the Materials and Methods section, all quantification was performed within the linear range of detection, before saturation of the signal.

Minor

1. There is no indication of which phosphorylation sites is assessed in western blotting figures.

As indicated in the Materials and Methods section, the anti-phospho-p70 S6 kinase recognizes p70 phosphorylated on Thr389. This information has been added to the figure legends.

Dear Dr. Kassel,

Thank you for the submission of your revised manuscript. We have now received the enclosed reports from the referees and I am happy to say that all support its publication now. Referees 2 and 3 still have a few minor suggestions that I would like you to address and incorporate before we can proceed with the official acceptance of your manuscript.

A few editorial requests will also need to be addressed:

- The Data Availability Section needs to be placed before the Acknowledgments
- The conflict of interest subheading needs to be corrected to Disclosure and Competing Interests Statement
- The author credits need to be removed from the ms file. All credits are entered during online ms submission.
- The statistics section in the author checklist has not been completed. Please send us a new, fully completed author checklist.
- This FUNDING INFO is missing in our online submission system: the KIT-Publication Fund of the Karlsruhe Institute of Technology. Please add it when you upload the final ms.
- A callout for Fig 4I is missing in the ms text, please add.
- The 2 EV tables should be uploaded as excel files and the table legends should be in the file, in the first cells.
- The Appendix file needs page numbers in the Table of content and throughout the Appendix file.
- Materials and Methods should be just 'Methods'

Figure Legends - Comments

- Please note that the legends for figure 8 is not provided in the sequential manner (legend for sub-figure D is provided before legend of sub-figure B). This needs to be rectified.

I would like to suggest one change to the title. Do you agree with :

Short internal open reading frames repress the translation of N-terminally truncated proteoforms

EMBO press papers are accompanied online by A) a short (1-2 sentences) summary of the findings and their significance, B) 2-3 bullet points highlighting key results and C) a synopsis image that is exactly 550 pixels wide and 200-600 pixels high (the height is variable). The synopsis image should provide a sketch of the major findings, like a graphical abstract. Please note that text needs to be readable at the final size. Please send us this information along with the final manuscript.

Kind regards,
Esther

Referee #1:

I would like to thank the authors for thoroughly addressing all my concerns and questions. I am impressed by the considerable effort and time they have dedicated to resolve the issues raised by me and the other reviewers. In my opinion, it has significantly strengthened the study, enabling more well-supported conclusions. Therefore, I am pleased to support its publication in EMBO reports.

Referee #2:

Overall, the authors have addressed the major concerns outlined in the first report with new experiments, analysis of relevant datasets, and rewording sentences to increase accuracy. There are some additional minor comments below that should be addressed.

1. In Figure 4B, it looks as if there may be an increase in nTRIP6/GR in the first (control, 0 IGF) and fourth (Everolimus, 0 IGF) sets of bars. Is this difference significant? If so, it might suggest that mTORC1's role on nTRIP6 expression isn't as simple as described.
2. On p. 10, there is a paragraph with a callout to "(Fig 2E, F)" that should be "(Fig 4F, G)" and a callout to "(Fig 4G, H)" that should be "(Fig 4H, I)".
3. In Figure 1F, "sort exposure" should be changed to "short exposure"

Referee #3:

The authors have largely addressed my concerns. One concern lingering is the possibility that mTOR induces alternative transcript variants giving rise to nTRIP6 without affecting TRIP6 total mRNA expression. Although this might seem farfetched, I would suggest adding a comment regarding this in the discussion.

Referee #1:

I would like to thank the authors for thoroughly addressing all my concerns and questions. I am impressed by the considerable effort and time they have dedicated to resolve the issues raised by me and the other reviewers. In my opinion, it has significantly strengthened the study, enabling more well-supported conclusions. Therefore, I am pleased to support its publication in EMBO reports.

We thank again the reviewer for the very constructive suggestions.

Referee #2:

Overall, the authors have addressed the major concerns outlined in the first report with new experiments, analysis of relevant datasets, and rewording sentences to increase accuracy. There are some additional minor comments below that should be addressed.

1. In Figure 4B, it looks as if there may be an increase in nTRIP6/GR in the first (control, 0 IGF) and fourth (Everolimus, 0 IGF) sets of bars. Is this difference significant? If so, it might suggest that mTORC1's role on nTRIP6 expression isn't as simple as described.

This difference is not statistically significant ($P=0.119$). We have added the P value on the graph in Figure 4B.

2. On p. 10, there is a paragraph with a callout to "(Fig 2E, F)" that should be "(Fig 4F, G)" and a callout to "(Fig 4G, H)" that should be "(Fig 4H, I)".

3. In Figure 1F, "sort exposure" should be changed to "short exposure"

We apologize for these mistakes which are now corrected.

Referee #3:

The authors have largely addressed my concerns. One concern lingering is the possibility that mTOR induces alternative transcript variants giving rise to nTRIP6 without affecting TRIP6 total mRNA expression. Although this might seem farfetched, I would suggest adding a comment regarding this in the discussion.

We have added this sentence to the discussion:

"Although unlikely, we cannot formally exclude the possibility that mTORC1 activity promotes the transcription of a *Trip6* mRNA variant that would code only for nTRIP6."

Dr. Olivier Kassel
Karlsruhe Institute of Technology (KIT)
Institute of Biological and Chemical Systems-Biological Information Processing (IBCS-BIP)
Hermann-von-Helmholtz Platz 1
Eggenstein-Leopoldshafen D-76344
Germany

Dear Dr. Kassel,

I am very pleased to accept your manuscript for publication in the next available issue of EMBO reports. Thank you for your contribution to our journal.
